# Towards Scalable and Robust Filtration Learning for Point Clouds via Principal Persistence Measure

**Kaifeng Zhang**                                                    *zhangkf2022@lamda.nju.edu.cn*
*State Key Laboratory for Novel Software Technology, Nanjing University, Nanjing, China*
*School of Artificial Intelligence, Nanjing University, Nanjing, China*

**Kai Ming Ting**                                                         *tingkm@nju.edu.cn*
*State Key Laboratory for Novel Software Technology, Nanjing University, Nanjing, China*
*School of Artificial Intelligence, Nanjing University, Nanjing, China*

**Reviewed on OpenReview:** *https://openreview.net/forum?id=8NkZxJFQ3I*

## Abstract

Topological features in persistent homology extracted via a filtration process have been shown to enhance the performance of machine learning tasks on point clouds. The performance is highly related to the choice of filtration, thereby underscoring the critical significance of filtration learning. However, the current supervised filtration learning method for point clouds cannot scale well. We identify that this shortcoming stems from the utilization of Persistence Diagrams (PD) for encoding topological features, such as connected components, rings or voids, etc. To address this issue, we propose to use Principal Persistence Measure (PPM), an existing statistical approximation of PD, as an alternative representation and adapt existing network for PPM-based filtration learning. Experimental results on point cloud classification tasks show that, in controlled filtration learning settings, the PPM-based framework preserves comparable classification performance in several cases while substantially improving the scalability for larger point clouds and degrading more gradually under outlier contamination.

## 1 Introduction

Topological information is shown to be effective in machine learning tasks on point clouds across many fields, such as biology (Kovacev-Nikolic et al., 2016; Liu et al., 2022; Meng et al., 2020; Xia et al., 2018; Xia & Wei, 2015) and chemistry (Hiraoka et al., 2016; Lee et al., 2017; Townsend et al., 2020). Topological features of a point cloud are built from a nested sequence of simplicial complexes (Salnikov et al., 2018) called *filtration*. The birth ($b_i \in \mathbb{R}$) and death ($d_i \in \mathbb{R}$) of a cycle, e.g., a ring (1-dimensional cycle) or a void (2-dimensional cycle), of the sequence are encoded in *Persistence Diagram* (PD) $\mathcal{D} = \{r_i = (b_i, d_i) \in \Omega | 1 \le i \le N(\mathcal{D})\}$, where **each $r \in \mathcal{D}$ is defined as a topological feature corresponding to the cycle**, $\Omega = \{(t_1, t_2) \in \mathbb{R}^2 | t_2 > t_1\}$ is an open half-plane and $N(\mathcal{D})$ is the number of topological features. PD can be transformed into a vector and fed into a machine learning model as input. A brief illustration of PD-based pipeline for point cloud [1] is shown in Figure 1(a,b,d,f).

In the pipeline, despite different learnable vectorization methods like Carrière et al. (2020); Kim et al. (2020); Reinauer et al. (2021), it is shown in Nishikawa et al. (2023) that the final performance is highly affected by the choice of the filtration. In addition, although there are different unsupervised filtration choices like Rips (Hausmann et al., 1995), DTM (Fasy et al., 2018) and Λ-filter (Zhang et al., 2023), supervised filtration

---

[1]Note that this pipeline can also be used for graph data, where the filtration usually employs unsupervised characteristics of a graph like degree (vertex-level), Ricci curvature (edge-level)(Ballester & Rieck, 2023; O'Bray et al., 2021; Southern et al., 2023) or outputs of supervised learnable networks (Hofer et al., 2020; Horn et al., 2022; Immonen et al., 2023; Mukherjee et al., 2024; Zhang et al., 2022). In this paper, we focus on the pipeline for point cloud.

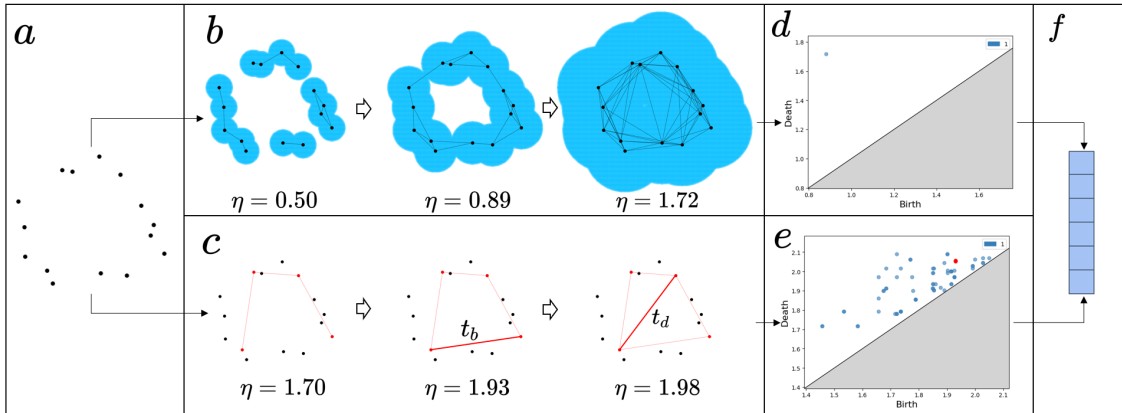

Figure 1: PD (a,b,d,f) and PPM-based (a,c,e,f) pipeline for point cloud. (a) Original point cloud $X$. Note that this can also be a distance matrix, since in the construction of simplicial complexes of the filtration, a pairwise connection is added if the pairwise distance is lower than a threshold $\eta$. This can be equivalently expressed that for balls centered at each point with radius of $\eta/2$, if two balls touch, an edge between two balls' centers is added. (b) An example filtration (Rips filtration) at values of $\eta$ of $X$. The union of balls is intended to show the process of the Rips filtration: when two balls touch (pairwise distance $\leq \eta$), an edge is added. According to Nerve theorem (Mona & Hintz, 2023), $X_\eta = \cup_{x \in X} B(x, \eta/2)$ is actually homotopy equivalent to Cěch complex. While the Rips filtration serves as a practical approximation of the Cěch filtration, it does not strictly reflect the topology of the union of balls. (c) An example filtration (Rips filtration) at various values of $\eta$ of one subset of $X$. The distances of the bold lines are $t_b$ and $t_d$. (d) 1-dimensional Persistence Diagram of $X$. (e) 1-dimensional Principal Persistence Measure of $X$. **The red point corresponds to the topological feature obtained via the red subset in (c).** (f) Vectorization of PD or PPM.

learning (Nishikawa et al., 2023) for point cloud often produces better results. We will refer this PD-based pipeline with filtration learning as PD-FL for short.

Based on weighted filtration, PD-FL (Nishikawa et al., 2023) develops a neural network architecture with isometry-invariance to learn the weight in an end-to-end way. However, since the computation algorithm [2] of filtration-induced PD is highly nontrivial to parallelize and can only be conducted by either pure CPU implementations (Bauer, 2021; Pérez et al., 2021) or a CPU-GPU hybrid one (AIDOS-Lab; Zhang et al., 2020), this network suffers from high time cost and cannot scale for a large point cloud.

In order to address this issue, we propose to replace PD in the pipeline with Principal Persistence Measure (PPM) (Gómez & Mémoli, 2024; Tung et al., 2025), which is a statistical approximation of PD, for the following two reasons: (1) As shown in Tung et al. (2025), PPM can be computed entirely on GPU in a parallel way, making it ideal for a scalable filtration learning framework. (2) PPM can capture topological information in a point cloud. PPM has been used in Tung et al. (2025) for latent space matching in Generative Adversarial Networks (Goodfellow et al., 2020). Although PPM is a rather vague approximation of PD and may not demonstrate all the possible topological features in a point cloud because the sampling mechanism favors points in dense region, the experiments in Figure 2 of Tung et al. (2025) demonstrate that a smaller distance (measured by Persistence Weighted Gaussian Kernel (Kusano et al., 2016) based Maximum Mean Discrepancy) between two PPMs indicates a smaller Wasserstein distance between the PDs. This shows that topological information can be obtained through PPM.

The main contributions of this work are threefold:

---

[2]As pointed by Zomorodian & Carlsson (2004), time complexity for computing PD in the worst case is $O(m^3)$, where $m$ is the number of simplices in the filtration. If we want topological feature corresponding to 1-dimensional cycle (ring), we need to consider up to 2-simplices in the complex. The number of simplices is $O(n^3)$ and the computational cost can be up to $O(n^9)$, where $n$ is the size of the point cloud.

1. Propose to use Principal Persistence Measure (PPM) in filtration learning framework (Nishikawa et al., 2023) to address the scalability limitation of PD-based approach and adapt existing framework for PPM-based Filtration Learning (PPM-FL).

2. Specialize existing EPD stability theory to the outlier-contamination model, obtaining a robustness bound for PPM in terms of the outlier ratio, and provide a controlled empirical robustness study.

3. Demonstrate that, across controlled filtration learning experiments, PPM-FL substantially improves the scalability of the topology-dependent training and preserves comparable classification performance in several settings. The accuracy effect is dependent on dataset and backbone: PPM-FL improves over PD-FL or over the non-topological representation in some settings, while PD-FL remains stronger in others. The main advantage of PPM-FL is therefore not uniform accuracy dominance, but a more scalable way to incorporate topological information into filtration learning for larger point clouds.

## 2 Background

We provide the relevant information about Persistence Diagram and weighted filtration (refer to Chazal & Michel (2021) for a comprehensive introduction).

### 2.1 Persistence Diagram

Let $\mathcal{X} \subset \mathbb{R}^d$ denote the ambient space, and let $X = \{x_1, \ldots, x_n\} \subset \mathcal{X}$ be a finite point cloud sampled from an underlying low-dimensional manifold $\mathcal{M} \subset \mathcal{X}$. Given a filtration function $g : \mathcal{X} \to \mathbb{R}_+$, the sublevel set at scale $\eta \geq 0$ is defined as

$$\mathcal{X}_\eta^g = \{x \in \mathcal{X} \mid g(x) \leq \eta\}.$$

For $\gamma \leq \eta$, these sublevel sets satisfy $\mathcal{X}_\gamma^g \subseteq \mathcal{X}_\eta^g$, yielding a filtration $\{\mathcal{X}_\eta^g\}_{\eta \geq 0}$. In point cloud applications, this filtration is constructed from the finite sample $X$, for example by using a Rips filtration (Hausmann et al., 1995) or a weighted filtration based on pairwise distances among points in $X$ (Nishikawa et al., 2023).

A topological feature is born at scale $b \in \mathbb{R}$ when it first appears in the filtration and dies at scale $d \in \mathbb{R}$ when it disappears. The pair $r = (b, d)$ is represented as a point in a persistence diagram

$$\mathcal{D} = \{r_i = (b_i, d_i) \in \Omega \mid 1 \leq i \leq N(\mathcal{D})\}, \qquad \Omega = \{(t_1, t_2) \in \mathbb{R}^2 \mid t_2 > t_1\},$$

where $N(\mathcal{D})$ is the number of topological features. An example using Rips filtration (Hausmann et al., 1995) $g(\cdot) = 2 \min_{x \in X} \ell(\cdot, x)$, where $\ell$ is Euclidean distance, and corresponding PD are provided in Figures 1(b) and 1(d).

The homology dimension is denoted by $q$: $q = 0$ corresponds to connected components, $q = 1$ to loops or rings, and $q = 2$ to voids. In practice, the filtration is usually truncated at a finite threshold, or the point with infinite death time is removed since it appears for all point clouds and does not provide discriminative information.

### 2.2 Weighted Filtration

In Rips filtration, at scale $\eta$, the sublevel set is $\mathcal{X}_\eta = \bigcup_{x \in X} B(x, \eta/2)$, where $B(x, \eta/2)$ is the ball centered at $x$ with radius $\eta/2$. Each ball has the same radius. Weighted filtration aims to put weight on each point's corresponding radius. For weighted filtration, at scale $\eta$, the sublevel set is $X_\eta = \bigcup_{x \in X} B(x, \eta - w(x))$, where $w(\cdot) : X \to \mathbb{R}$ is the weight function. Current work (Nishikawa et al., 2023), i.e., PD-FL, designed a network to learn this weight function $w$. In order to align with the setting in PD-FL and ensure a fair comparison, our work is also based on the weighted filtration.

## 3 Related Work

### 3.1 Filtration Learning

Supervised filtration learning is first introduced for graph data [3] by designing learnable vertex filter function (Hofer et al., 2020). It is then studied for more extensive and general purposes on graph data (Horn et al., 2022; Immonen et al., 2023; Mukherjee et al., 2024; Zhang et al., 2022). Filtration learning for point cloud is rarely developed. Existing work (Nishikawa et al., 2023) built a network via weighted filtration, i.e., given a point cloud $X \subset \mathbb{R}^d$, one can define the radius value $r_x(\eta)$ at scale $\eta$ for $x \in X$ as $r_x(\eta) = \eta - w(x)$ if $\eta > w(x)$, otherwise $r_x(\eta) = -\infty$, where $w$ is the weight function. DTM (Fasy et al., 2018) is a special case of this weighted filtration where $w$ is distance-to-measure function.

It is required in Nishikawa et al. (2023) that for filtration learning, the weight function $w(\cdot) = f(X, \cdot) : \mathbb{R}^d \to \mathbb{R}$ needs to meet the following three conditions:

1. $f$ should be determined by the whole point cloud $X$ and does not depend on the order of points in $X$;

2. $f$ should be isometry-invariant, i.e, for any isometric transformation $T$, any point cloud $X$, and $x \in X$, $f(TX, Tx) = f(X, x)$;

3. The output of the $f(X, x)$ should have both of global information $X$ and pointwise information of $x$.

A network [4] that relies on deepsets (Zaheer et al., 2017) and takes distance matrix as input is designed to satisfy the conditions above. Once the filtration is determined by this network, PD of the entire point cloud is computed and then can be vectorized as input to a Multi-Layer Perceptron for classification task. Although this network outperforms unsupervised filtration like Rips and DTM, it relies on CPU for the computation of PD and the substantial time cost (Zomorodian & Carlsson, 2004) for computing PD prevents its scalability.

### 3.2 Principal Persistence Measure

One way to reduce the time cost of computing PD of the entire point cloud is to use statistical approximation: for multiple random subsets [5] (with fixed size) of the entire point cloud $X$, a PD $\mathcal{D}_i = \{r_j = (b_j, d_j) \in \Omega | 1 \leq j \leq N(\mathcal{D}_i)\}$ is obtained for each random subset $X_s^i \subset X$. The statistical approximation, i.e., Expected PD (EPD) (Chazal & Divol, 2018), takes a distributional view and represents each PD of subset $X_s^i$ as a measure $\mu_i = \sum_{j=1}^{N(\mathcal{D}_i)} \delta_{r_j}$ supported on $\Omega$ where $\delta_{r_j}$ is Dirac point mass at $r_j$. The empirical EPD is then the average $\bar{\mu} = \frac{1}{M} \sum_{i=1}^{M} \mu_i$, where $M$ is the number of random subsets.

Principal Persistence Measure (PPM) (Gómez & Mémoli, 2024) is a special case of Expected PD where each subset $X_s^i$ has fixed size $2q + 2$ where $q$ is the homology dimension ($q = 0$ for connected component, $q = 1$ for rings, etc.). It is guaranteed that each PD on $2q + 2$ points has at most one single topological feature $r_1 = (t_b, t_d)$ when $q \geq 1$, i.e., $|N(\mathcal{D}_i)| \leq 1$, which can be efficiently computed as follows: given any $x \in X_s^i$, let $x^{(1)}, x^{(2)} \in X_s^i$ be the points such that $d(x, x^{(1)}) \geq d(x, x^{(2)}) \geq d(x, a)$ for any $a \in X_s^i \setminus \{x^{(1)}, x^{(2)}\}$ where $d$ is a distance function (filtration), then

$$r_1 = (\max_{x \in X_s^i} d(x, x^{(2)}), \min_{x \in X_s^i} d(x, x^{(1)})).$$

PPMs have stability with respect to Wasserstein distance, as shown in Theorem A.2. The computation of PPM can be easily implemented in a parallel way and conducted on GPU and the existence of $r_1$ is discussed in Theorem A.3. By considering random subsets, the computation of PPM costs less time than computing PD on the entire point cloud.

In order to improve scalability, we propose to use PPM, instead of PD, to encode topological information and adapt existing network (Nishikawa et al., 2023) for PPM-based filtration learning, which learns from multiple subsets.

---

[3] Filtration Learning on graph data is scalable due to the simplicity of computing PD on graph. In the filtration, the addition of an edge either connects two connected components or creates a ring.

[4] The full architecture of network (Nishikawa et al., 2023) is shown in Appendix A.5.

[5] Each point in the subset is i.i.d. sampled from the entire point cloud.

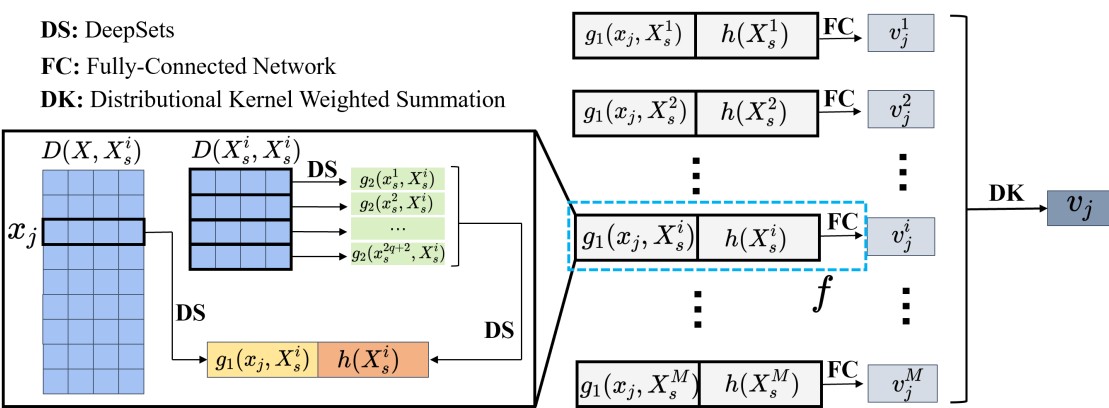

Figure 2: PPM-based Filtration Learning Framework. Value $v_j$ is the output of $\bar{f}(\mathbb{X}_s, x_j)$. $D$ represents a distance matrix for two point clouds $X$ and $Y$, $D(X,Y) = (d(x_i, y_j))_{i=1,j=1}^{i=|X|,j=|Y|} \in \mathbb{R}^{|X| \times |Y|}$, where $d$ is Euclidean distance.

## 4    Filtration Learning for PPM

We propose the filtration learning framework for PPM (PPM-FL) based on a weighted filtration. PPM-FL, which concentrates on learning from multiple subsets, is adapted from the network in Nishikawa et al. (2023) for learning from the entire point cloud. In PPM-FL, for each sampled subset $X_s$, the network first produces a subset-dependent weight contribution for point $x_j$. The final filtration weight of $x_j$ is obtained by aggregating these subset-dependent contributions.

Different from the three conditions mentioned in Nishikawa et al. (2023), here weight function $w$ should be related to $\mathbb{X}_s = \cup_{i \in [M]} \{X_s^i\}$, with each sampled subset $X_s^i \subset X$ of size $2q + 2$, where $q$ is the homology dimension, instead of the whole point cloud $X$. This gives the following three adapted requirements for the weight function $w(\cdot) = \bar{f}(\mathbb{X}_s, \cdot)$:

1. The output of $\bar{f}(\mathbb{X}_s, x)$ should be determined by each subset $X_s^i$ and does not depend on the order of the points. The closer a subset $X_s^i$ is to $x$, the greater its impact on $x$ should be.

2. $\bar{f}$ should be isometry-invariant, i.e., for any isometric transformation $T$, any point cloud $X$, $\mathbb{X}_s$ and $x \in X$, $\bar{f}(T\mathbb{X}_s, Tx) = \bar{f}(\mathbb{X}_s, x)$, where $T\mathbb{X}_s \triangleq \cup_{i \in [M]} \{TX_s^i\}$.

3. The output of $\bar{f}(\mathbb{X}_s, x)$ should have both of global information $\mathbb{X}_s$ and pointwise information of $x$.

In order to deal with set $\mathbb{X}_s = \cup_{i \in [M]} \{X_s^i\}$, we model the weight function $\bar{f}$ in a simple summation form, i.e., $\bar{f}(\mathbb{X}_s, \cdot) = \sum_{i=1}^M K(X_s^i, \cdot) f(X_s^i, \cdot)$, where $f$ is used to output the weight $(v_j^i)$ of the j-th point $x_j$ in $X$ w.r.t. subset $X_s^i$, i.e., $v_j^i = f(X_s^i, x_j)$. Function $K$ measures the similarity between the input and subset $X_s^i$ to meet requirement 1: the closer $X_s^i$ is to $x$, the greater its impact on $x$. The filtration learning framework for PPM, as shown in Figure 2, is then adapted from Nishikawa et al. (2023) to meet the three new requirements:

1. Following Nishikawa et al. (2023), the independence on the order of points is guaranteed by following DeepSets (Zaheer et al., 2017) architecture $g_1$ (the feature extractor of the distances between a point $x_j$ and random subset $X_s^i$), $g_2$ and $h$ ( feature extractor of random subset $X_s^i$) :

$$g_1(x_j, X_s^i) = \phi^{(2)}(\mathbf{op}(\{\phi^{(1)}(d(x, x_j)) | x \in X_s^i\}));$$

$$g_2(x_s^k, X_s^i) = \phi^{(4)}(\mathbf{op}(\{\phi^{(3)}(d(x, x_s^k)) | x \in X_s^i\}));$$

$$h(X_s^i) = \phi^{(5)}(\mathbf{op}(\{g_2(x, X_s^i) | x \in X_s^i\})),$$

where $x_s^k$ is the k-th [6] point in a subset $X_s^i$, $d$ is Euclidean distance, $\phi^{(i)}$s are all fully-connected neural networks and **op** is the permutation invariant operator.

For each $x_j \in X$, after the weight $v_j^i$ w.r.t. the $i$-th subset $X_s^i$ is obtained, the final weight value $v_j$ is the distributional kernel weighted summation of each $v_j^i$, i.e., $v_j = \sum_{i=1}^M K(x_j, X_s^i)v_j^i$, where $K(x_j, X_s^i) = \frac{1}{|X_s^i|}\sum_{x \in X_s^i} \kappa(x, x_j)$ and $\kappa(x, x_j) = \exp(\frac{-\|x-x_j\|_2^2}{2\sigma^2})$. $K(x_j, X_s^i)$ is a locality-biased kernel assigning larger influence to sampled subsets closer to $x_j$. This locality weighting should be understood as an inductive bias, analogous to kernel smoothing, rather than as a necessary mathematical property of PPM. It preserves the desired permutation and isometry invariance while encouraging nearby sampled subsets to have greater influence on the learned filtration value.

2. The isometry-invariance is guaranteed by using the distance matrix as input.

3. The global and pairwise information are stored in $h$ and $g_1$ respectively.

Once all the weights are obtained, the weighted filtration is used to compute the PPM of the point cloud [7]. The space cost of PPM-FL is $O((2q+2)nM + (2q+2)^2 M)$, where $q$ is the homology dimension, $M$ is the number of subsets in PPM and $n$ is the point cloud size.

**Proposition 4.1** (Time Complexity Comparison). *Let $n$ denote the point cloud size, $M$ the number of random subsets, $q$ the homology dimension, and $t_\phi$ the time for a single forward pass of a fully-connected network $\phi^{(i)}$. Then:*

1. *The per-sample time complexity of PPM-FL is $O\big(M(2q+2)(n+2q+2)t_\phi + M(2q+2)^2\big)$. The first term accounts for the network computing $M$ weight values for each of the $n$ points (via distance matrices of size $(2q+2) \times n$ and $(2q+2) \times (2q+2)$), and the second term accounts for computing the PPM topological feature for each subset.*

2. *The per-sample time complexity of PD-FL is $O\big(n^2 t_\phi + T_{\mathrm{PD}}(n,q)\big)$, where $T_{\mathrm{PD}}(n,q)$ denotes the cost of computing the $q$-dimensional persistence diagram. Let $m$ denote the number of simplices involved in the filtration. The standard matrix reduction algorithm for persistent homology has worst-case time complexity $O(m^3)$ (Zomorodian & Carlsson, 2004). For $q$-dimensional homology on a point cloud with $n$ points, the filtration must include simplices up to dimension $q+1$. Since a $(q+1)$-simplex is determined by $q+2$ vertices, the number of relevant simplices is bounded by $m = O(n^{q+2})$. Substituting this into the $O(m^3)$ persistence-computation bound gives $T_{\mathrm{PD}}(n,q) = O(m^3) = O\big((n^{q+2})^3\big) = O\big(n^{3(q+2)}\big)$.*

The PPM computation instead uses $M$ random subsets of fixed size $2q+2$, which avoids constructing and reducing the full filtration on all $n$ points: since $2q+2$ is a small constant (e.g., 4 for $q=1$), the PPM-FL complexity is effectively $O(Mnt_\phi)$, which scales linearly in $n$ for fixed $M$. In contrast, PD-FL scales at least as $O(n^2 t_\phi)$ for the network and up to $O(n^9)$ for computing 1-dimensional PD.

*Remark* 4.2 (End-to-End GPU Training). A practical advantage of PPM-FL beyond time complexity is that the entire forward and backward pass runs on GPU. In PD-FL, the PD computation and its gradient (AIDOS-Lab) require CPU–GPU data transfer at every training iteration, creating a communication bottleneck. In PPM-FL, the topological feature $r_1 = (t_b, t_d)$ of each subset is computed via differentiable max and min operations on pairwise distances, so gradients flow through the PPM computation natively on GPU without any CPU involvement.

As for the approximation ability, for weight function $\bar{f}(\mathbb{X}_s, \cdot) = \sum_{i=1}^M K(X_s^i, \cdot)f(X_s^i, \cdot)$, function $f$ is able to approximate any continuous function, as shown in Theorem 4.1 of Nishikawa et al. (2023).

---

[6]This index k is used to differentiate two different point in subset $X_s^i$. We just assume a random order here since the framework is permutation invariant.

[7]Note that we could use the framework in Nishikawa et al. (2023), which uses the distance matrix of the entire point cloud to learn the weights. But the space cost would be $O(n^2)$, which is one order of magnitude higher than PPM-FL's $O((2q+2)nM + (2q+2)^2 M)$.

PPM can be then vectorized [8] by a supervised method PersLay (Carrière et al., 2020) for a potential task. For example, if the task is point cloud classification, we can input the resulting vector into MLP that employs a cross entropy loss function. Due to the parallel implementation of PPM, this PPM-based pipeline can be deployed on GPU to scale on a large point cloud.

## 5    Robustness of PPM against Outliers

Here we demonstrate the robustness of PPM against outliers. We start by considering the robustness of a general case of PPM, i.e., Expected Persistence Diagram (EPD), where the size of random subset $X_s$ is denoted as $n$, and then extend the result to PPM. Let $\mathcal{M}$ denote the underlying manifold of point cloud $X$; $P$ denote the density of the distribution on $k$-dimensional $\mathcal{M}$; and $U$ denote the density of the distribution of outliers, such as a uniform distribution. Following the setting in Vishwanath et al. (2020); Cai et al. (2025), the outlier-contaminated distribution $P_o$ is expressed as

$$P_o = (1 - \epsilon) \cdot P + \epsilon \cdot U,$$

where $\epsilon$ is the percentage of outliers. $X$ is a sample of $P_o$. Then we have the following lemma on the robustness of EPD under the assumption that filtration $\mathcal{K}$ meets K1-K5 requirements in Section 3 of Chazal & Divol (2018).

We have the following assumptions on $P$ and $U$:

1. We assume that $P$ and $U$ share the same support. Regarding the manifold's support, $U$ is a (maybe uniform) distribution over the entire support of $\mathcal{M}$, including regions where the inlier density $P$ are very low and points have little chance to be sampled from this low density area. An example is shown in Figure 16 in Gómez & Mémoli (2024). The support of $P$ is the circular area and the low density region lies at the inner part of the ring.

2. $P$ and $U$ are both lower and upper bounded in the support, i.e. there exist constants $\bar{c} \geq \underline{c} > 0$ such that $\bar{c} \geq P(x) \geq \underline{c}$ and $\bar{c} \geq U(x) \geq \underline{c}$ for any $x$ in the support. This assumption is identical to assumption 2(ii) in Cai et al. (2025), which shares the same contamination model $P_o = (1 - \epsilon) \cdot P + \epsilon \cdot U$.

**Lemma 5.1.** *For EPD $\mathbb{E}_{X_s \sim P^n}[\nu(X_s)]$ with density $\mu_1$ and $\mathbb{E}_{X_s \sim P_o^n}[\nu(X_s)]$ with density $\mu_2$, where $\nu(X_s)$ is the measure form of PD $\mathcal{D}(\mathcal{K}(X_s))$ , i.e., $\nu(X_s) = \sum_{r \in \mathcal{D}(\mathcal{K}(X_s))} \delta_r$, it holds that*

$$\|\mu_1 - \mu_2\|_1 \leq C_n H_k(\mathcal{M})^n p_{n-1}(\epsilon)\epsilon,$$

*where $C_n$ is the expected number of points in the PD built with the filtration $\mathcal{K}$ on $n$ i.i.d. points on $\mathcal{M}$, $H_k(\mathcal{M})$ is $k$-dimensional Hausdorff measure [9] of $\mathcal{M}$ and $p_{n-1}(\epsilon)$ is a polynomial of order $n-1$ with bounded coefficients.*

The proof is provided in Appendix B.

**Theorem 5.2.** *For PPM $\mathbb{E}_{X_s \sim P^{2q+2}}[\nu(X_s)]$ with density $\mu_1$ and $\mathbb{E}_{X_s \sim P_o^{2q+2}}[\nu(X_s)]$ with density $\mu_2$, it holds that*

$$\|\mu_1 - \mu_2\|_1 \leq H_k(\mathcal{M})^{2q+2} p_{2q+1}(\epsilon)\epsilon,$$

*where $H_k(\mathcal{M})$ is $k$-dimensional Hausdorff measure of $\mathcal{M}$ and $p_{2q+1}(\epsilon)$ is a polynomial of order $2q + 1$ with bounded coefficients.*

---

[8]Despite many methods for vectorization for PD or PPM, including supervised (Carrière et al., 2020; Kim et al., 2020; Reinauer et al., 2021) and unsupervised ones (Adams et al., 2017; Bubenik et al., 2015; Chung & Lawson, 2022), we choose to use PersLay (Carrière et al., 2020) , a supervised vectorization that uses similar structure like Deepsets for simplicity.

[9]Hausdorff measure is a generalization of the traditional notions of area and volume to non-integer dimensions. Let $k$ be a non-negative integer. For $A \subset \mathcal{M}$, and $\delta > 0$, consider $H_k^\delta(A) = \inf\{\sum_i \alpha(k)(\frac{\text{diam}(U_i)}{2})^k, A \subset \bigcup_i U_i \text{ and } \text{diam}(U_i) < \delta\}$, where $\alpha(k)$ is the volume of the $k$-dimensional unit ball and *diam* represents diameter. The $k$-dimensional Hausdorff measure on $\mathcal{M}$ of $A$ is defined by $H_k(A) = \lim_{\delta \to 0} H_k^\delta(A)$.

*Proof.* Following Lemma 5.1, PPM is a special case of EPD where the size of each subset is $n = 2q + 2$, $q$ is homology dimension. Combined with the fact that each subset of size $2q + 2$ has at most 1 topological feature, i.e., $C_n \leq 1$, we can have the result above. $\qquad\square$

Theorem 5.2 indicates that a small number of outliers in the point cloud will not severely disturb PPM. Theorem 5.2 should be viewed as a specialization of existing stability results for Expected Persistence Diagrams (EPDs), particularly the framework of Divol & Lacombe (2021b), rather than as an entirely independent stability theory. Our setting focuses on the specific outlier-contamination model $P_o = (1 - \epsilon)P + \epsilon U$. Under this model, Theorem 5.2 expresses the perturbation of PPM in terms of $\epsilon$, yielding a polynomial dependence on the outlier ratio for the PPM subset size $2q + 2$. The role of the theorem in this paper is therefore to adapt the EPD stability perspective to the contamination model used in our robustness experiments and to connect this bound with the PPM-FL setting.

The relation between the upper bound and outlier distribution $U$ is discussed in Appendix B.3. The experimental demonstration of PPM's robustness with the learned filtration in point cloud classification task is provided in Section 6.2. We further discuss the relation between Theorem 5.2 and existing results in Divol & Lacombe (2021b) in Appendix D.4.

## 6 Experiments

We compare PPM-FL with PD-FL [10] on the protein (Kovacev-Nikolic et al., 2016) and ModelNet10 (Wu et al., 2015) datasets used previously in similar evaluations (Nishikawa et al., 2023). We extend to ModelNet40 in Appendix D.6. We also conduct an ablation study to compare PPM-FL with the unsupervised Rips filtration to demonstrate the effectiveness of PPM-FL. PPM-based approach cannot use DTM since there are only $2q+2$ points in each random subset. In addition, we validate the robustness and scalability of PPM-FL. In summary, under controlled filtration learning experiments, PPM-FL preserves comparable classification performance in several settings while substantially improving the scalability of the topology-dependent second phase for larger point clouds. In the robustness experiment, PPM-FL degrades more gradually than PD-FL at moderate and high outlier ratios.

All the classification results are obtained through 3-fold cross validation. All the experiments are conducted on an Ubuntu 20.04 system with 2TB RAM, AMD EPYC 7763 64-Core 1500 MHZ CPU and NVIDIA A6000 GPU. The network is implemented with PyTorch 2.0.1. The code implementation of PD-FL is from `https://github.com/git-westriver/FiltrationLearningForPointClouds`. The details of the experiment setting are provided in Appendix C. Code is provided at `https://github.com/CaptainJuice/PPM-FL`.

The purpose of the experiments is not to compete with specialized modern point cloud recognition architectures on absolute ModelNet accuracy. Instead, our goal is to evaluate whether PPM can replace PD inside the filtration learning pipeline while preserving comparable task performance and improving scalability. For this reason, our primary baseline is PD-FL under the controlled protocol of Nishikawa et al. (2023), rather than full-scale 3D classifiers optimized for sota recognition accuracy. The reported ModelNet accuracies should therefore be interpreted as controlled evidence for the PD-to-PPM replacement, not as a claim of sota 3D recognition performance.

### 6.1 Comparison with PD-FL

Following the same setting in Nishikawa et al. (2023) on the protein dataset, we employ the pipeline shown in Figure 1 with filtration learning, which uses the topological information only for classification. The results are shown in Table 1. The accuracies of PPM-FL over different homology dimensions are higher than PPM-

---

[10]Here PD-FL means the PD-based Filtration Learning (Nishikawa et al., 2023). The experiments on comparing PD-based filtration learning (PD-FL) with Rips (PD-Rips) and DTM (PD-DTM) filtration for PD have already been conducted in Nishikawa et al. (2023). PD-Rips produces similar results to that of PD-DTM. And PD-FL outperforms PD-Rips and PD-DTM. Hence, we just use PD-FL as baseline in our work.

Rips [11] and the standard deviation of PPM-FL is lower. This comparison with PPM-Rips demonstrates the advantage of the supervised filtration learning over the unsupervised filtration like Rips.

Table 1: Accuracy of the binary classification task of protein structure. We compared our method PPM-FL with PPM-Rips and PD-FL. PPM-Rips denotes using the Rips filtration instead of the filtration learning for PPM. The PersLay vectorization is learned in an end-to-end way. Homology dimension $q$ stands for the dimension of PDs (PPMs). $q = 0\&1$ means that we use both 0-dimensional and 1-dimensional PD or PPM, and the feature input to the MLP is the concatenation of the vectorization of 0-dimensional and 1-dimensional PDs (PPMs) via PersLay.

|          | $q = 0$          | $q = 1$          | $q = 0\&1$       |
|----------|------------------|------------------|------------------|
| PD-FL    | $65.80 \pm 1.76$ | $82.10 \pm 1.66$ | $81.70 \pm 1.31$ |
| PPM-Rips | $64.09 \pm 7.78$ | $65.40 \pm 2.12$ | $71.50 \pm 1.31$ |
| PPM-FL   | $84.00 \pm 1.39$ | $80.20 \pm 1.76$ | $84.60 \pm 1.22$ |

On the protein dataset, the comparison is dependent on the homology dimension. For $q = 0$, PPM-FL substantially outperforms PD-FL, achieving $84.00 \pm 1.39\%$ compared with $65.80 \pm 1.76\%$. For $q = 0\&1$, PPM-FL also performs better, while for $q = 1$, PD-FL remains stronger. This suggests that PPM-FL should not be viewed as always merely matching PD-FL, nor as uniformly dominating it: PPM-FL preserves or improves accuracy in certain settings. The large improvement of PPM-FL over PD-FL at $q = 0$ may come from that PPM aggregates many randomly sampled pairwise observations. Since 0-dimensional persistent homology is closely related to the edge weights of the minimum spanning tree, this aggregation may act as a regularized summary of pairwise connectivity on noisy protein distance matrices. In contrast, for $q = 1$, exact PD computation over the full point cloud may better preserve precise ring structures. We present this as an interpretation of the observed results rather than a proved mechanism.

Overall, both approaches are effective in extracting meaningful topological features from the protein dataset for classification, suggesting that the key aspect for achieving good performance in this context is not solely the precise topological information but rather the ability to appropriately integrate topological information with the classification model.

For the ModelNet10 dataset, we use the version which contains 10 classes, with 100 instances in each class. Each instance is a point cloud of shape $128 \times 3$. Here we employ the same two-phase process in Nishikawa et al. (2023) by combining topological embedding with a DNN-based method. For a point cloud $X$, let $\Psi_{\text{topo}}(X) \in \mathbb{R}^{L_1}$ be the topological embeddings from PD-FL or PPM-FL, $\Psi_{\text{DNN}}(X) \in \mathbb{R}^{L_2}$ be the feature from a DNN-based method. Let $\ell$ be the loss function and $m$ be the number of classes. The two-phase training classifiers proposed by Nishikawa et al. (2023) are specified as follows:

1. Phase 1 classifier receives feature from $\Psi_{\text{DNN}}$ as $C_1 : \mathbb{R}^{L_2} \to \mathbb{R}^m$, where $m$ is the number of classes. $C_1$ and $\Psi_{\text{DNN}}$ are jointly learned by minimizing $\sum_j \ell(C_1(\Psi_{\textbf{DNN}}(X_j), y_j)$.

2. Phase 2 classifier $C_2 : \mathbb{R}^{L_1 + L_2} \to \mathbb{R}^m$. We fix the parameters of the learned $\Psi_{\text{DNN}}$ in the first phase and learn $C_2$ and $\Psi_{\text{topo}}$ by minimizing $\sum_j \ell(C_2([\Psi_{\text{DNN}}(X_j)^\intercal, \Psi_{\text{topo}}(X_j)^\intercal]^\intercal), y_j)$.

The final classification is conducted through $C_2$ with the concatenated features from $\Psi_{\text{DNN}}$ and $\Psi_{\text{topo}}$ on the test set. For the choices of DNN method, we consider DeepSets (Zaheer et al., 2017), PointNet (Qi et al., 2017) and PointMLP (Ma et al., 2022). The results of the two-phase process are shown in Table 2.

When using DeepSets in the first phase, PD-FL, PPM-Rips and PPM-FL can all improve the classification accuracy in the second phase. Except for the case of $q = 1$ where PD-FL outperforms PPM-Rips and PPM-FL, PPM-FL achieves comparable results with PD-FL and PPM-Rips. Note that the differences between PPM-FL and PD-FL are within standard deviations in most settings. This demonstrates the benefit of

---

[11] In the original PD-FL work (Nishikawa et al., 2023), it has been demonstrated that the Rips and DTM yield comparable results when applied to both the protein and the ModelNet10 dataset. For the sake of simplicity and to streamline our analysis, we will solely compare our proposed method with Rips.

Table 2: Accuracies for the classification task on the ModelNet10 dataset. Two-phase training process is utilized here. The first phase we use DeepSets and PointNet. The results of PointMLP are discussed in Appendix D.1. The first phase directly uses point cloud as input and does not have the notion of homology dimension since it does not involve topological summary like PD or PPM. So there is no result for each homology dimension in the first phase.

| 1st Phase | | 2nd Phase | | |
|---|---|---|---|---|
| DeepSets | | PD-FL | PPM-Rips | PPM-FL |
| | $q = 0$ | $67.40 \pm 2.31$ | $67.30 \pm 2.41$ | $67.90 \pm 3.01$ |
| $66.23 \pm 3.19$ | $q = 1$ | $68.20 \pm 2.37$ | $67.50 \pm 2.17$ | $66.90 \pm 2.17$ |
| | $q = 0\&1$ | $67.40 \pm 0.82$ | $67.10 \pm 1.42$ | $67.50 \pm 2.88$ |
| PointNet | | PD-FL | PPM-Rips | PPM-FL |
| | $q = 0$ | $68.60 \pm 3.09$ | $67.10 \pm 2.21$ | $68.70 \pm 2.23$ |
| $67.23 \pm 1.80$ | $q = 1$ | $68.90 \pm 2.64$ | $67.10 \pm 2.21$ | $69.60 \pm 1.33$ |
| | $q = 0\&1$ | $69.00 \pm 5.57$ | $67.30 \pm 2.02$ | $69.80 \pm 0.77$ |

topological information when combined with the DNN method, and suggests that PPM captures sufficient topological signal for classification despite being an approximation.

When using PointNet in the first phase, PPM-Rips makes no improvement on the classification accuracy. PPM-FL produces comparable results to that of PD-FL: both of them improve the accuracy in the second phase. The highest accuracy ($69.80 \pm 0.77$) is obtained via PointNet + PPM-FL with homology dimensions $q = 0\&1$.

The additional results of PointMLP are discussed in Appendix D.1. The ablation study of the Gaussian weight is provided in Appendix D.3. As shown in Table 7, the two weighting schemes (Gaussian and Uniform) lead to similar accuracies under the tested ModelNet10 settings, with differences generally within standard deviations. We therefore use the Gaussian kernel as a simple locality-biased aggregation rule, rather than as a design choice whose statistical superiority is established by this ablation.

It is worth noting that using all the homology dimensions is a generally reliable choice to get high accuracy. Hence we will use all the homology dimensions in the following experiments.

**Note on statistical significance.** The accuracy differences between PPM-FL and PD-FL on ModelNet10 are generally within standard deviations (e.g., $67.50 \pm 2.88$ vs. $67.40 \pm 0.82$ for DeepSets with $q = 0\&1$), which is expected given the 3-fold cross-validation on a relatively small dataset (1000 instances). We emphasize that the primary contribution of PPM-FL is not to surpass PD-FL in accuracy but to achieve comparable classification performance while providing substantially better scalability for large point clouds and improved robustness against outliers.

## 6.2 Robustness

Here we show the robustness of PPM against outliers in the point cloud. For the ModelNet10 dataset, we use the model trained on outlier-free point clouds; and for each point cloud in the test set, we add outliers from uniform distribution at percentage $\epsilon$. We choose the two-phase process where the first phase uses DeepSets and the second phase uses all the homology dimensions $q = 0\&1$, since in this case PPM-FL and PD-FL produces similar results when there is no outliers on the test set, ensuring a fair comparison.

The average accuracies of PPM-FL and PD-FL under different outlier percentages $\epsilon$ are shown in Figure 3. PPM-FL and PD-FL behave similarly at small outlier ratios ($\epsilon < 8\%$), while the gap becomes clearer as the outlier ratio increases. PPM-FL degrades more gradually overall and maintains accuracy above 55% up to $\epsilon \leq 18.5\%$. We interpret this as empirical support for the robustness motivation of PPM.

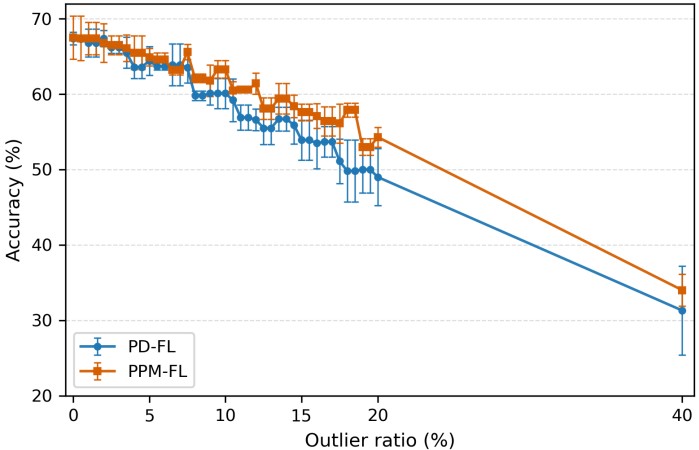

Figure 3: Average classification accuracies and corresponding standard deviations under different outlier ratios ($\epsilon$). Detailed analysis is provided in Appendix D.5.

## 6.3 Scalability w.r.t. Point Cloud Size

We demonstrate the scalability of PPM-FL w.r.t. the point cloud size $n$. The settings in each phase here are the same as that in last subsection on robustness. The number of the point clouds in the dataset is fixed. We consider both homology dimensions 0 and 1 in the second phase. We fix the batch size and report the average [12] of each epoch in the second phase in Table 3. In the first phase, we use the same pretrained model for PPM-FL and PD-FL.

Table 3: Actual time cost (s) per-epoch under different numbers $n$ of points in the point cloud and numbers of random subsets $M$ in PPM. Our aim is to compare the real-world usage of these methods on the ModelNet10 dataset. Thus, PPM-FL is conducted on GPU and PD-FL is on pure CPU or CPU-GPU Hybrid. The computation of PD-FL (CPU) is through the GUDHI library (Maria et al., 2014). The computation of PD-FL (CPU-GPU Hybrid) is through the *torch-topological* library (AIDOS-Lab). We stopped the run of PD-FL (CPU) for $n = 1024$ after the per-epoch runtime exceeded 10 hours; therefore, the exact wall-clock time is not reported.

| $n$ | 1st Phase DeepSets (GPU) | PPM-FL (GPU) $M = 100$ | $M = 200$ | $M = 400$ | PD-FL (CPU) | PD-FL (Hybrid) |
|---|---|---|---|---|---|---|
| 64 | $0.31_{\pm 0.08}$ | $56.61_{\pm 1.08}$ | $112.79_{\pm 1.41}$ | $210.17_{\pm 2.50}$ | $41.46_{\pm 2.55}$ | $5.59_{\pm 0.23}$ |
| 128 | $0.34_{\pm 0.06}$ | $58.91_{\pm 3.11}$ | $118.79_{\pm 1.22}$ | $218.26_{\pm 1.35}$ | $187.56_{\pm 2.87}$ | $29.46_{\pm 2.42}$ |
| 256 | $0.32_{\pm 0.04}$ | $65.85_{\pm 1.04}$ | $129.19_{\pm 1.15}$ | $239.42_{\pm 1.01}$ | $1101.85_{\pm 11.46}$ | $157.95_{\pm 1.95}$ |
| 512 | $0.37_{\pm 0.07}$ | $77.95_{\pm 1.77}$ | $154.32_{\pm 1.58}$ | $273.47_{\pm 1.26}$ | $7556.32_{\pm 66.11}$ | $607.95_{\pm 13.20}$ |
| 1024 | $0.54_{\pm 0.08}$ | $91.32_{\pm 1.74}$ | $136.84_{\pm 0.73}$ | $278.68_{\pm 6.23}$ | $> 10h$ | $7020.36_{\pm 750.85}$ |

The 1st Phase DeepSets runtime is below one second per epoch across all tested point cloud sizes, ranging from $0.31 \pm 0.08$s at $n = 64$ to $0.54 \pm 0.08$s at $n = 1024$. Thus, the computational bottleneck is the topology-dependent second phase. For small point clouds, especially $n \leq 128$, the CPU-GPU hybrid PD-FL implementation can be faster than PPM-FL because PPM-FL processes many random subsets. As $n$ increases, however, the cost of PD-FL grows rapidly, while PPM-FL remains practical for fixed $M$. At $n = 1024$, PD-FL (CPU) exceeded 10 hours per epoch and PD-FL (Hybrid) required $7020.36 \pm 750.85$s per epoch, whereas PPM-FL required $91.32 \pm 1.74$s, $136.84 \pm 0.73$s, and $278.68 \pm 6.23$s for $M = 100, 200, 400$,

---

[12] All timing results are wall-clock measurements on the available hardware. Since some GPU measurements were collected on a shared server, small non-monotonic fluctuations can occur due to scheduling and resource contention. The overall trend across $n$ and the contrast with PD-FL remain unchanged.

respectively. When the number of points in each point cloud $n$ is fixed, the time cost of PPM-FL is linear w.r.t. $M$.

**When to prefer PPM-FL over PD-FL.** We note that for small point clouds ($n \leq 128$), PD-FL (Hybrid) is faster than PPM-FL due to the overhead of processing $M$ subsets. The crossover occurs around $n = 256$, where PPM-FL (GPU, $M = 100$) reaches parity with PD-FL (Hybrid). For $n \geq 512$, PPM-FL is clearly faster. In practice, PPM-FL is recommended when point clouds are large or when GPU-only computation is desired. For small point clouds where a CPU-GPU hybrid setup is available, PD-FL (Hybrid) remains a competitive option.

Table 4: Accuracies of PPM-FL (GPU) under different $M$s ($n = 128, q = 0\&1$) on the ModelNet10 dataset with the 1st phase model being DeepSets.

| $M = 25$ | $M = 50$ | $M = 100$ | $M = 200$ | $M = 400$ |
|---|---|---|---|---|
| $67.00 \pm 2.53$ | $67.00 \pm 1.43$ | $67.60 \pm 2.20$ | $67.50 \pm 2.88$ | $67.40 \pm 1.92$ |

Table 5: Accuracies of PPM-FL (GPU) under different $n$s ($q = 0\&1$, $M = 100$ for $n \leq 512$, $M = 200$ for $n = 1024$) on the ModelNet10 dataset with the 1st phase model being DeepSets.

| $n = 128$ | $n = 256$ | $n = 512$ | $n = 1024$ |
|---|---|---|---|
| $67.60 \pm 2.20$ | $71.65 \pm 1.84$ | $69.40 \pm 2.94$ | $71.50 \pm 1.72$ |

Tables 4 and 5 report accuracy results for selected scalable configurations corresponding to the timing study in Table 3. These results are intended as accuracy sanity checks rather than as a complete statistical characterization of the optimal $M$ or $n$. Within the tested range, PPM-FL maintains comparable accuracy while substantially reducing the topology-dependent runtime reported in Table 3. In particular, the $n = 1024$ ModelNet10 result in Table 5 achieves $71.50 \pm 1.72\%$, which is consistent with the smaller point cloud settings ($67.60 \pm 2.20\%$ at $n = 128$, $71.65 \pm 1.84\%$ at $n = 256$, and $69.40 \pm 2.94\%$ at $n = 512$). Thus, the improved runtime at larger $n$ is not obtained by an obvious collapse in ModelNet10 classification accuracy.

The $M$-ablation in Table 4 suggests that performance is relatively stable across the tested numbers of random subsets, but it does not establish a precise degradation threshold. In practice, $M$ should be treated as a computational-accuracy trade-off parameter: larger $M$ uses more sampled subsets and increases runtime, while smaller $M$ gives a cheaper approximation.

## 7 Discussion

### 7.1 On absolute accuracy

We note that the absolute classification accuracies on ModelNet10 ($\sim$67–70%) are modest compared to state-of-the-art 3D point cloud methods that use full-scale architectures. However, following the setting of Nishikawa et al. (2023), the experimental setup uses only 100 instances per class with 128 points each, which is a deliberately small-data regime designed to evaluate the contribution of topological features rather than compete with large-scale 3D recognition methods. In this regime, PPM-FL often achieves accuracy comparable to PD-FL while being more scalable in the larger-$n$ settings.

### 7.2 On higher homology dimensions

Although our experiments focus on $q = 0$ and $q = 1$, the PPM construction can in principle be applied to $q \geq 2$. For example, when $q = 2$, each PPM subset has size $2q + 2 = 6$, so the per-subset computation remains small and GPU-parallelizable. Thus, from a computational perspective, $q = 2$ is feasible.

However, the practical usefulness of $q = 2$ features on ModelNet point clouds is less clear. A 2-dimensional homology class corresponds to void-like or enclosed surface structures, which may appear for sufficiently dense

samples of watertight CAD objects. In practice, ModelNet meshes are heterogeneous and are not guaranteed to be watertight; many sampled point clouds contain open surfaces, thin structures, or incomplete cavities. As a result, $q = 2$ features can be less stable and more sensitive to sampling density than $q = 0$ and $q = 1$. In addition, the robustness bound in Theorem 5.2 becomes weaker as $q$ increases because the subset size $2q + 2$ and the polynomial order in the outlier ratio both grow with $q$.

For these reasons, we focus our empirical study on $q = 0$ and $q = 1$, which are more consistently present and commonly used in point-cloud topology. Extending PPM-FL to stable $q = 2$ features on denser or explicitly watertight 3D datasets is an interesting direction for future work.

## 8   Conclusion

In this study, we propose to use PPM to replace PD in a filtration learning framework. PPM-based filtration learning (PPM-FL) addresses the scalability limitations of existing PD-based approach for point clouds. By leveraging PPM, which can be computed entirely on GPU in a parallel manner, we achieved a more efficient solution for encoding topological features.

Our theoretical analysis specializes existing EPD stability results to an outlier-contamination model for PPM. The robustness experiment is consistent with this motivation: PPM-FL degrades more gradually than PD-FL when the test point cloud is contaminated with outliers.

**Limitation.** The approximation nature of PPM may lead to the loss of some fine-grained topological information. Future work could explore ways to enhance the representational power of PPM while maintaining its computational efficiency. In addition, Theorem 5.2 on robustness has practical limitations, as its dependence on $q$ makes it weak for higher homology dimension. This leads to the result that the fraction of noise has to be very small to control the measure errors. Our experimental results in Figure 3 align with this: while PPM-FL is not highly robust, it degrades more gradually than PD-FL as outliers increase (maintaining accuracy above 55% when $18.5\% \geq \epsilon$), showing a modest improvement.

### Broader Impact Statement

This work introduces a scalable and robust framework for filtration learning on point clouds using Principal Persistence Measure. By enabling efficient topological data analysis on larger datasets and demonstrating robustness to outliers, our method has potential applications in diverse fields ranging from bioinformatics (e.g., analyzing protein structures) to computer vision (e.g., 3D object classification). We are not aware of any specific negative societal impacts that would directly stem from this work.

### Acknowledgments

This work is supported by National Natural Science Foundation of China under Grants No. W2531050 and 92470116.

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

# A   Background

## A.1   Expected Persistence Diagram

A PD $\mathcal{D} = \{r_i = (b_i, d_i) \in \Omega | 1 \le i \le N(\mathcal{D})\}$ can be equivalently represented as a counting measure $\mu$ on $\Omega$ given by $A \in \mathcal{B} \to \mu(A) = \sum_{i=1}^{N(\mathcal{D})} \delta_{r_i}(A)$,

where $\mathcal{B}$ is the class of all Borel subsets of $\Omega$ and $\delta_r$ denotes the Dirac point mass at $r \in \Omega$. When each sampled PD is a random draw from a distribution $P$, its EPD, denoted as $\mathbb{E}[\mu]$, is defined as $A \in \mathcal{B} \to \mathbb{E}[\mu](A) = \mathbb{E}[\mu(A)]$ (Chazal & Divol, 2018).

Given a finite set $\{\mu_1, \mu_2, ..., \mu_n\}$, consisting of sampled PDs from $P$, the empirical EPD is defined as $\bar{\mu} = \frac{1}{n}\sum_{i=1}^{n}\mu_i$. The support of $\bar{\mu}$ is $\mathcal{S}_{\bar{\mu}} = \cup_{i=1}^{n}\mathcal{D}_i$, where $\mathcal{D}_i = \{r_j = (b_j, d_j) \in \Omega | 1 \le j \le N(\mathcal{D}_i)\}$ is the support of the sampled PD $\mu_i$. EPD can be viewed as a distribution (Chazal & Divol, 2018) of topological features supported on the open half plane $\Omega$. Quantization method (Divol & Lacombe, 2021a) has been developed to reduce the support size of EPD.

An inherent metric applicable to the space of EPDs (Divol & Lacombe, 2021a) with the same mass, is Wasserstein distance (Villani, 2009). In a space of EPDs with different masses, the Optimal Partial Transport metric ($\text{OT}_p$) (Figalli, 2010) is used to allow any mass transportation from or to the diagonal $\partial\Omega$.

For the approximation error of EPD, we have the following theorem in Cao & Monod (2022):

**Theorem A.1** (Cao & Monod (2022)). *Let $X \subset \mathbb{R}^m$ be a finite set of points, and $\pi$ be a probability measure on $X$ satisfying the $(a, b, r_0)$-standard assumption. Suppose $X_s^1, \ldots, X_s^M$ are $M$ i.i.d. samples from the distribution $\pi^{\otimes n}$ with $|X_s^i| = n$. Let $\bar{\mu}$ be the empirical persistence measure, i.e., EPD; denote $\beta := \frac{p}{b} - 1$. Then the empirical persistence measure approaches the true persistence measure $\mathcal{D}(X)$ (here we omit the symbol filtration choice and $\mathcal{D}(X)$ is in measure form: $\mathcal{D}(X) = \sum_r \delta_r$ ) of $X$ in expectation at the following rates:*

$$\mathbb{E}[\text{OT}_p^p(\bar{\mu}, D(X))] \leq \begin{cases} O(M^{-1/2}) + O(1) + O\left(n^{-\beta}\right) & \text{if } p > b; \\ O(M^{-1/2}) + O(1) + O\left(\left(\frac{\log n}{n}\right)^{1/b}\right) & \text{if } p \leq b, r_0 < \left(\frac{\log n}{an}\right)^{1/b}; \\ O(M^{-1/2}) + O(1) + O\left(\left(\frac{\log n}{n}\right)^{p/b} \frac{1}{(\log n)^2}\right) & \text{if } p \leq b, r_0 \geq \left(\frac{\log n}{an}\right)^{1/b}. \end{cases}$$

## A.2 Principal Persistence Measure

As a special case of Expected Persistence Diagram, the approximation error and convergence analysis of the empirical measure to Principal Persistence Measure is theoretically studied in Theorem 3.20 in Gómez & Mémoli (2024): the empirical measure converges to PPM almost surely as the number of subsets $M \to \infty$.

For the stability of PPM, we have the following Theorem A.2 from Tung et al. (2025).

**Theorem A.2** (Tung et al. (2025)). *Let $p \geq 1$, and let $W_p$ denote the $p$-Wasserstein metric on $\mathbb{R}^d$ and $\Omega'$. A key property shown in Gómez & Mémoli (2024) Theorem 3.8, Theorem 4.11 is that PPMs are stable:*

$$W_p(PPM_q(\mu), PPM_q(\nu)) \leq C_q W_q(\mu, \nu),$$

*for all $\mu, \nu \in \mathcal{P}_c(\mathbb{R}^d)$, where $C_q > 0$ is a constant which depends on homology dimension $q$ and $\mathcal{P}_c(\mathbb{R}^d)$ is the Borel probability measure with compact support on $\mathbb{R}^d$.*

The $PPM_q(\mu)$ here means the Principal Persistence Measure with homology dimension $q$ computed from points sampled from measure $\mu$ supported on $\mathbb{R}^n$ and the PPM is transformed from the (birth, death) to (birth, persistence) space $\Omega' = \{(b, l) \in \mathbb{R}^2\} \setminus \{l = 0\}$, where persistence=death-birth $\geq 0$. This is a slightly rotated version of the PPM we considered in the $\Omega = \{(t_1, t_2) \in \mathbb{R}^2 | t_2 > t_1\}$.

For the computation of PPM, we have the following theorem:

**Theorem A.3** (Gómez & Mémoli (2024)). *Let $(X, d_X)$ be a metric space with $n$ points. Then:*

1. *For all homology dimension larger than $\frac{n}{2} - 1$, the PD obtained via Rips filtration is empty.*

2. *If $n$ is even and homology dimension equals to $\frac{n}{2} - 1$, then the PD obtained via Rips filtration consists of a single point $r_1 = (t_b, t_d)$ if and only if $t_b < t_d$, and is empty otherwise.*

## A.3 Notes on Death time

Death times are finite when the homology dimension is zero and this results a point $r = (0, \infty)$ in the PD, representing the final connected component. In practice, for vectorization, we usually set the maximum threshold for the filtration to be a finite value or just remove the point with infinite death time since it exists for all point clouds and has no practical value. So here we focus on the practice aspect and assume the death time is always finite. We ignore the topological feature with infinite death time.

## A.4 Vectorization via PersLay

PersLay (Carrière et al., 2020) is a supervised vectorization method for PD, a multiset $\mathcal{D} = \{r_i = (b_i, d_i) \in \Omega | 1 \leq i \leq N(\mathcal{D})\}$ on the half plane $\Omega = \{(t_1, t_2) \in \mathbb{R}^2 | t_2 > t_1\}$. PersLay is adapted from the DeepSets structure (Zaheer et al., 2017) and expressed as follows:

$$\text{PersLay}(\mathcal{D}) = \mathbf{op}(\{w(r) \cdot \phi(r)\}_{r \in \mathcal{D}}),$$

where **op** is any permutation invariant operation (such as minimum, maximum, sum, kth largest value...), $w : \mathbb{R}^2 \to \mathbb{R}$ is a weight function for the points in PD, and $\phi : \mathbb{R}^2 \to \mathbb{R}^q$ is a representation function called point transformation, mapping each point $(b_i, d_i)$ of a PD to a vector.

Certain choices of $w$, $\phi$ and **op** can transform PersLay into specific methods like Persistence Image (Adams et al., 2017) and Persistence Landscape (Bubenik et al., 2015). In our work, we follow the setting in Nishikawa et al. (2023) and choose $w(\cdot) = 1$ and $\phi$ as

$$\phi(r) = [\exp(-\frac{\|p - c_1\|^2}{2}), \exp(-\frac{\|p - c_2\|^2}{2}), ..., \exp(-\frac{\|p - c_m\|^2}{2})]^\mathsf{T},$$

where $r = (b, d)$, $p = (b, d - b)$ and all the $c_i$s are the parameters to be learned.

**Non-DL vectorization.** While our current work is more centered on the scalability of our proposed method within the DL-based context, we recognize the importance of the non-DL perspective. Regarding the non-DL based vectorization method, the scalability issue of Expected Persistence Diagram (EPD), a more general form of PPM, has been discussed in detail in Bubenik et al. (2015) and Section 4.3.1 in Gómez & Mémoli (2024).

### A.5 Persistence Diagram-based Filtration Learning Framework

The PD-based filtration learning framework (PD-FL) (Nishikawa et al., 2023) is shown in Figure 4. PD-FL tries to learn the weight function $w(\cdot) = f_\theta(X, \cdot)$ from the entire point cloud $X$ for weighted filtration. Once the weight is learned, weighted filtration is used to compute PD. Then, PD is vectorized by PersLay and used in machine learning task like point cloud classification.

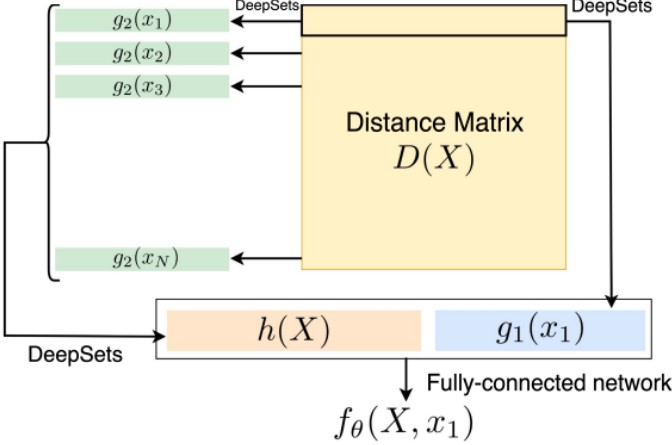

Figure 4: Persistence Diagram-based Filtration Learning Framework. Functions $g_1$, $g_2$ and $h$ are the same as those we use in Section 4. This is a direct reuse of Figure 2 in Nishikawa et al. (2023).

## B Proof and Related Analysis

### B.1 On the assumption of $P$ and $U$ in Lemma 5.1 & Theorem 5.2

We have the following two assumptions on $P$ and $U$.

1. We assume that $P$ and $U$ share the same support. Regarding the manifold's support, $U$ is a (maybe uniform) distribution over the entire support of $\mathcal{M}$, including regions where the inlier density $P$ are very low and points have little chance to be sampled from this low density area. An example is shown in Figure 16 in Gómez & Mémoli (2024). The support of $P$ is the circular area and the low density region lies at the inner part of the ring.

2. $P$ and $U$ are both lower and upper bounded in the support, i.e. there exist constants $\bar{c} \geq \underline{c} > 0$ such that $\bar{c} \geq P(x) \geq \underline{c}$ and $\bar{c} \geq U(x) \geq \underline{c}$ for any $x$ in the support.

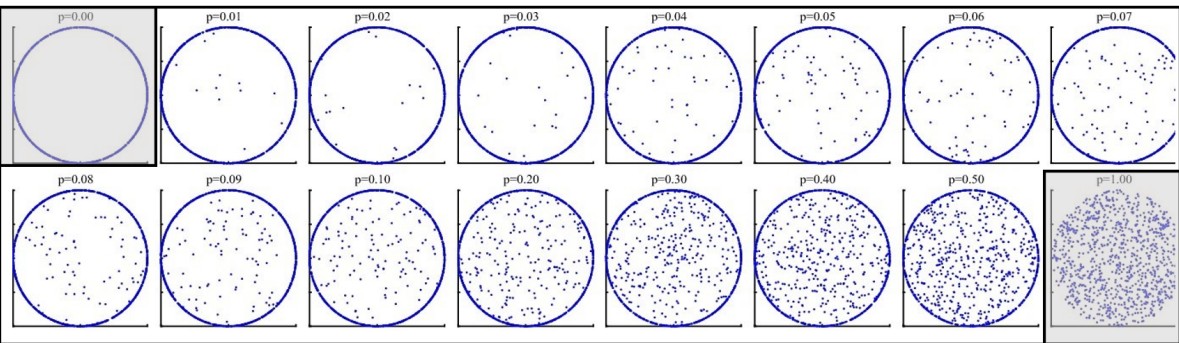

Figure 5: Part of Figure 16 from Gómez & Mémoli (2024).

The caption of Figure 16 in Gómez & Mémoli (2024) claims that "Given $0 \leq p \leq 1$, we sample $X \subset D^2$ with 1000 points so that each point is uniformly distributed in the interior of $D^2$ with probability $p$ or on its boundary $\mathbb{S}^1_E$ with probability $1-p$". We can consider $P$ is such a distribution with a $p$ that is close to 0, but it is not 0. And consider $U$ is a distribution with $p$ that is close to 1, but it is not 1. So both the supports of $P$ and $U$ are $D^2 \cup \mathbb{S}^1_E$ and the value of their pdf is upper and lower bounded. And it would not cause the pdf being either 0 or infinite, as shown in the area not shaded in Figure 5. This aligns with our assumptions above.

## B.2 Proof of Lemma 5.1

**Lemma 5.1.** For EPD $\mathbb{E}_{X_s \sim P^n}[\nu(X_s)]$ with density $\mu_1$ and $\mathbb{E}_{X_s \sim P_o^n}[\nu(X_s)]$ with density $\mu_2$, where $\nu(X_s)$ is the measure form of PD $\mathcal{D}(\mathcal{K}(X_s))$, i.e., $\nu(X_s) = \sum_{r \in \mathcal{D}(\mathcal{K}(X_s))} \delta_r$, it holds that

$$\|\mu_1 - \mu_2\|_1 \leq C_n H_k(\mathcal{M})^n p_{n-1}(\epsilon)\epsilon,$$

where $C_n$ is the expected number of points in the PD built with the filtration $\mathcal{K}$ on $n$ i.i.d. uniform points on $\mathcal{M}$, $H_k(\mathcal{M})$ is $k$-dimensional Hausdorff measure of $\mathcal{M}$ and $p_{n-1}(\epsilon)$ is a polynomial of order $n-1$ with bounded coefficients.

*Proof.* For Theorem 7.1 in Chazal & Divol (2018), we have that

$$\|\mu_1 - \mu_2\|_1 \leq C_n H_k(\mathcal{M})^n \|q_1 - q_2\|_\infty,$$

where $q_1(q_2)$ is the density with respect to the Hausdorff measure $H_k$, $q_1(x) = \Pi_{i=1}^n P(x_i)$ and $q_2(x) = \Pi_{i=1}^n P_o(x_i)$, where $x = [x_1^\mathsf{T}, ..., x_n^\mathsf{T}]^\mathsf{T}$. For the $\|q_1 - q_2\|_\infty$ term, since each $x_i$ is i.i.d. sampled, for $x = \text{argmax}|q_1 - q_2|$, it holds that

$$(q_1 - q_2)(x) = \Pi_i P(x_i) - \Pi_i P_o(x_i)$$
$$= \Pi_i P(x_i)[1 - \frac{\Pi_i P_o(x_i)}{\Pi_i P(x_i)}],$$

where under the assumption that each $x_i$ is in the support (with positive density) of $P$ and $1 > \frac{U(x_i)}{P(x_i)} \geq \Delta > 0$, we have that

$$\frac{\Pi_i P_o(x_i)}{\Pi_i P(x_i)} = \Pi_i[1 - (1 - \frac{U(x_i)}{P(x_i)})\epsilon]$$
$$\geq \Pi_i[1 - (1 - \Delta)\epsilon]$$

So it holds that

$$\|q_1 - q_2\|_\infty = |q_1(x) - q_2(x)|$$
$$\leq \Pi_i P(x_i)[1 - (1 - (1 - \Delta)\epsilon)^n]$$
$$= p_{n-1}(\epsilon)\epsilon.$$

Finally, we obtain that

$$\|\mu_1 - \mu_2\|_1 \leq C_n H_k(\mathcal{M})^n p_{n-1}(\epsilon)\epsilon.$$

$\square$

### B.3 The dependence on outlier distribution $U$

The upper bound in Lemma 5.1 is dependent on $U$. This dependence is reflected in the coefficients of the polynomial $p_{n-1}(\epsilon)$. As shown in the proof above, the polynomial's coefficients are derived from $\Delta = \inf \frac{U(x)}{P(x)}$ (the minimum ratio of outlier to inlier densities over $\mathcal{M}$), which is dependent on noise $U$.

For the area where $P$ is dense and $U$ is sparse, a sparser $U$ will result a small $\Delta$. This will lead to smaller absolute values of the coefficients in the polynomial $p_{n-1}(\epsilon)$, i.e. a smaller upper bound. This corresponds to the intuition that EPD (or PPM) is more robust to a sparser outlier distribution.

## C Experiment Settings

**Optimization.** We set the batch size as 128 for protein datasets and 40 for ModelNet10 dataset. For optimizer, we use Adam. Learning rate is 0.1 for ModelNet and 0.001 for protein dataset. For scheduler, we use the TransformerLRScheduler implemented in pytorch (`https://github.com/sooftware/pytorch-lr-scheduler/tree/main`) and the number of warm-up epoch is set to be 40. For ModelNet10, the number of epochs in the first phase is 1500. For protein dataset and the second phase of ModelNet dataset, we use the EarlyStopping handler (`https://pytorch.org/ignite/generated/ignite.handlers.early_stopping.EarlyStopping.html`) with patience=20/10 for ModelNet/Protein datasets and min_delta $= 0.002$ for the loss on validation set of size 200.

**Principal Persistence Measure.** For the computation of PPM, we set $M = 200$ for both protein and ModelNet10 datasets. For the vectorization method PersLay, the length of vectorization $m$ is set to be 32 and the permutation invariant operation **op** is summation.

**Networks.** For the DNN-based methods (DeepSets, PointNet and PointMLP) in the first phase, we use the same structure as those in Nishikawa et al. (2023). We set all of the permutation invariant operators **op** that appear in PPM-FL and PD-FL are all summation. The dimension of the feature vectors obtained by PersLay is set as 16, except that when using both homology dimensions, it is 32. The DeepSets-like structures $\phi^{(1)} - \phi^{(5)}$ and fully connected network are the same as those in Nishikawa et al. (2023). We initialized the parameters in PPM-FL with normal distribution with a mean of 0 and a standard deviation of 1.0. Other parameters were initialized with the default settings of PyTorch.

**Datasets.** The protein dataset (Kovacev-Nikolic et al., 2016) does not provide a point cloud. Instead, a cross-correlation matrix $\mathbb{C}$ is provided for each protein. Then the dynamic distance matrix $\mathbb{D}$, where $\mathbb{D}_{i,j} = 1 - |\mathbb{C}_{i,j}|$, is used to compute PD or PPM. We use a version of this dataset (Nishikawa et al., 2023), which contains two classes of protein, with 500 instances in each class. Each instance is a distance matrix of shape $60 \times 60$ and has noise from a uniform distribution with standard deviation of 0.1 for the off-diagonal elements. For the ModelNet10 dataset, we use the version which contains 10 classes, with 100 instances in each class. Each instance is a point cloud of shape $128 \times 3$.

## D Additional Results and Discussions

### D.1 Results of two-phase training (PointMLP + PPM-FL)

Nishikawa et al. (2023) observed that applying PD-FL after PointMLP reduces the accuracy of the 1st phase model. We include the corresponding PPM-FL result to examine whether the same issue occurs for the PPM-based second phase.

The PointMLP results in Table 6 show a limitation of the current two-phase topology module. Unlike DeepSets and PointNet, PointMLP already uses local neighborhood construction and pairwise geometric

Table 6: Accuracies for the classification task of ModelNet10 dataset when the first phase is PointMLP.

| 1st Phase | PointMLP | | $70.10 \pm 4.70$ | |
|---|---|---|---|---|
| | | $q = 0$ | $q = 1$ | $q = 0\&1$ |
| 2nd Phase | PPM-Rips | $54.20 \pm 9.56$ | $57.30 \pm 13.37$ | $49.29 \pm 11.82$ |
| | PPM-FL | $52.70 \pm 8.58$ | $53.90 \pm 8.23$ | $56.59 \pm 13.01$ |

information in its backbone. Adding the current PPM-FL or PD-FL second phase after PointMLP substantially degrades performance, which suggests a harmful interaction or incompatibility rather than a useful complementary topological signal. We therefore do not treat PointMLP as positive evidence for PPM-FL. Instead, this result indicates that the current PPM-FL design is better suited to simpler first-stage backbones such as DeepSets and PointNet, and that integrating PPM-style topological features with stronger local geometric backbones requires further study.

## D.2 Relation between PPM and the distribution of pairwise distances.

It is worth mentioning that when homology dimension $q = 0$ and we use the unsupervised Rips filtration instead of the filtration learning, PPM represents the distribution of pairwise distances, which is shown in Boutin & Kemper (2005) to almost solve the isometry classification problem for point clouds. PPM of higher homology dimension can be viewed as a conditional distribution of pairwise distances that are topologically meaningful for high dimensional cycles.

## D.3 Ablation study on weight choice: Gaussian or Uniform

We present supplementary experimental results of PPM-FL with Gaussian ($\sum_{i=1}^{M} K(X_s^i) f(X_s^i, \cdot)$) or Uniform ($\sum_{i=1}^{M} f(X_s^i, \cdot)$) weight on the ModelNet10 dataset, under the same setting as that in Table 2. In the first phase, we use DeepSets. The results are shown in Table 7. For the choice of $K$, any differentiable distributional kernel would be fine. For simplicity, we choose to use Gaussian Distribution Kernel. The hyperparameter $\sigma$ is set to be the median of all the pairwise distances.

Table 7: Accuracies of PPM-FL on ModelNet10 dataset under different weight functions.

| | $q = 0$ | $q = 1$ | $q = 0\&1$ |
|---|---|---|---|
| Gaussian | $67.90 \pm 3.01$ | $66.90 \pm 2.17$ | $67.50 \pm 2.88$ |
| Uniform | $67.60 \pm 1.97$ | $67.00 \pm 2.21$ | $66.80 \pm 1.87$ |

In our pipeline, the first phase uses DeepSets, while the second phase employs PPM-FL with either a Gaussian or Uniform weight function. As shown in Table 7, PPM-FL with Gaussian weighting and PPM-FL with uniform weighting achieve similar accuracies across the tested homology dimensions. Because the differences are small and generally within standard deviations, Table 7 should not be interpreted as demonstrating statistically significant superiority of the Gaussian kernel. The result instead suggests that the method is not highly sensitive to this particular weighting choice in the tested ModelNet10 setting. We retain the Gaussian kernel as a locality-biased aggregation rule.

## D.4 Relation between Theorem 5.2 and existing results in Divol & Lacombe (2021b).

Here we discuss the difference between our theoretical results (Theorem 5.2) with the two Propositions (5.4 & 5.5) in Section 5.3 of Divol & Lacombe (2021b).

- In Proposition 5.4, the support of distribution $P$ and $P'$ is $\mathcal{M}^p$, the space of PDs (measures) with finite persistence. Proposition 5.4 demonstrates that the expectation of $P$, i.e. EPD, is stable with respect to the distortion ($P'$) of $P$. While our result is about the stability of EPD with respect

to the addition of outliers of distribution $P$, which is supported on $\mathbb{R}^d$ (instead of $\mathcal{M}^p$), the space where we sample the subsets.

- Proposition 5.5 demonstrates the stability of EPD with respect to the distribution $\xi$ supported on $\mathbb{R}^d$. This is similar to our result in Lemma 5.1, with $P(P_o)$ corresponding to $\xi(\xi')$. Compared with Proposition 5.5, our result (Lemma 5.1) takes a specific form of $P_o = (1 - \epsilon) \cdot P + \epsilon \cdot U$ (a mixture of the original distribution $P$ and outlier distribution $U$) and links the upper bound to the mixture proportion $\epsilon$, while Proposition 5.5 generally gives the upper bound as the bottleneck distance $W_\infty(\xi, \xi')$). It could be argued that our result is a specific case of Proposition 5.5 of Divol & Lacombe (2021b).

### D.5 Results under different outlier percentages

Table 8: Accuracies and standard deviations under different outlier percentages ($\epsilon$). The first phase uses DeepSets. The second phase uses both homology dimensions $q = 0\&1$.

| $\epsilon$ | PPM-FL | PD-FL | $\epsilon$ | PPM-FL | PD-FL |
|---|---|---|---|---|---|
| 0.0% | $67.50 \pm 2.88$ | $67.40 \pm 0.82$ | 10.5% | $60.50 \pm 1.16$ | $59.20 \pm 2.81$ |
| 0.5% | $67.40 \pm 2.94$ | $67.30 \pm 0.29$ | 11.0% | $60.60 \pm 0.09$ | $56.90 \pm 1.65$ |
| 1.0% | $67.40 \pm 2.13$ | $66.80 \pm 1.84$ | 11.5% | $60.60 \pm 0.09$ | $56.90 \pm 1.65$ |
| 1.5% | $67.40 \pm 2.13$ | $66.80 \pm 1.84$ | 12.0% | $61.40 \pm 1.42$ | $56.60 \pm 1.42$ |
| 2.0% | $66.80 \pm 2.55$ | $67.40 \pm 1.07$ | 12.5% | $58.10 \pm 1.45$ | $55.50 \pm 2.16$ |
| 2.5% | $66.50 \pm 1.25$ | $66.20 \pm 0.76$ | 13.0% | $58.10 \pm 1.45$ | $55.50 \pm 2.16$ |
| 3.0% | $66.50 \pm 1.25$ | $66.20 \pm 0.76$ | 13.5% | $59.40 \pm 2.02$ | $56.70 \pm 1.59$ |
| 3.5% | $66.10 \pm 1.76$ | $65.50 \pm 2.05$ | 14.0% | $59.40 \pm 2.02$ | $56.70 \pm 1.59$ |
| 4.0% | $65.50 \pm 2.22$ | $63.60 \pm 1.53$ | 14.5% | $58.40 \pm 1.49$ | $55.90 \pm 2.53$ |
| 4.5% | $65.50 \pm 2.22$ | $63.60 \pm 1.53$ | 15.0% | $57.60 \pm 1.11$ | $53.90 \pm 2.66$ |
| 5.0% | $64.90 \pm 1.16$ | $64.40 \pm 1.91$ | 15.5% | $57.60 \pm 1.11$ | $53.90 \pm 2.66$ |
| 5.5% | $64.60 \pm 0.90$ | $63.70 \pm 0.49$ | 16.0% | $57.10 \pm 1.62$ | $53.50 \pm 3.36$ |
| 6.0% | $64.60 \pm 0.90$ | $63.70 \pm 0.49$ | 16.5% | $56.40 \pm 1.96$ | $53.70 \pm 2.02$ |
| 6.5% | $63.20 \pm 0.61$ | $63.90 \pm 2.75$ | 17.0% | $56.40 \pm 1.96$ | $53.70 \pm 2.02$ |
| 7.0% | $63.20 \pm 0.61$ | $63.90 \pm 2.75$ | 17.5% | $56.10 \pm 2.59$ | $51.10 \pm 2.94$ |
| 7.5% | $65.60 \pm 1.03$ | $63.50 \pm 2.03$ | 18.0% | $57.90 \pm 0.91$ | $49.80 \pm 4.12$ |
| 8.0% | $62.10 \pm 0.57$ | $59.80 \pm 0.61$ | 18.5% | $57.90 \pm 0.91$ | $49.80 \pm 4.12$ |
| 8.5% | $62.10 \pm 0.57$ | $59.80 \pm 0.61$ | 19.0% | $53.00 \pm 1.09$ | $50.00 \pm 3.09$ |
| 9.0% | $61.80 \pm 2.09$ | $60.10 \pm 1.52$ | 19.5% | $53.00 \pm 1.09$ | $50.00 \pm 3.09$ |
| 9.5% | $63.30 \pm 1.15$ | $60.10 \pm 2.00$ | 20.0% | $54.30 \pm 1.32$ | $49.00 \pm 3.79$ |
| 10.0% | $63.30 \pm 1.15$ | $60.10 \pm 2.00$ | 40.0% | $34.00 \pm 2.11$ | $31.30 \pm 5.91$ |

We report accuracies and standard deviations corresponding to Figure 3 in Table 8. Around $\epsilon = 0$, the accuracies of PPM-FL and PD-FL drop with a similar rate. When $\epsilon \leq 1.5\%$, the accuracy of PPM-FL almost remains the same while PD-FL drops from 67.40 to 66.80. This demonstrates that a few outliers have more impact on PD-FL than PPM-FL. PD-FL does not ignore the first outliers while PPM-FL does. The reason behind this may be that when the number of outliers are small, the outliers have very little chance to be selected in a small subset. When $\epsilon = 40.0\%$, both PPM-FL and PD-FL have very bad performance.

Some adjacent outlier ratio settings in Table 8 have identical reported accuracies and standard deviations. This is not caused by reusing the final statistics across different experiments. In the robustness experiment, the outlier ratio $\epsilon$ is converted into an integer number of inserted outlier points for each test point cloud. Therefore, nearby values of $\epsilon$ can correspond to the same effective number of inserted outliers after integer rounding. In addition, the same random seed is used across settings to make the comparison between PPM-FL and PD-FL controlled. As a result, adjacent outlier-ratio settings may occasionally evaluate identical corrupted point clouds and thus produce identical aggregate statistics. We retain these entries in the table to report the full sweep over the nominal outlier ratios.

### D.6 Results on ModelNet40 dataset

Table 9: Test accuracy (%) on ModelNet40 (3-fold cross validation, $n = 1024$, homology dimension $q = 0\&1$).

| Method | CV0 | CV1 | CV2 | Mean | Std |
|---|---|---|---|---|---|
| DeepSets (1st Phase) | 62.22 | 64.52 | 63.54 | 63.43 | 0.94 |
| PPM-FL (2nd Phase) | 64.47 | 66.17 | 63.99 | **64.88** | 0.93 |
| $\Delta$ | +2.25 | +1.65 | +0.45 | +1.45 | — |

On the ModelNet40 dataset (Wu et al., 2015), we evaluate the two-phase framework against the first-stage DeepSets model under identical data splits. The standalone DeepSets model achieves $63.43 \pm 0.94\%$ across three cross validation folds, while the two-phase PPM-FL model achieves $64.88 \pm 0.93\%$. The absolute accuracy is not intended to compete with specialized sota ModelNet40 architectures. Rather, this experiment serves as an additional feasibility check on a more challenging benchmark with $n = 1024$ points. The consistent numerical improvement across the three folds suggests that the PPM-FL second phase can provide useful additional signal under this controlled protocol, but we do not interpret this result as a sota ModelNet40 comparison or as conclusive evidence of a statistically significant accuracy gain.

