# OpenReview forum: "Towards Scalable and Robust Filtration Learning for Point Clouds via Principal Persistence Measure"
_TMLR — Accepted by TMLR_

### Review · Reviewer_XvTB · 2026-05-05

**Summary Of Contributions:**

This paper addresses the scalability bottleneck of filtration learning for point clouds (PD‑FL), which relies on computing persistence diagrams (PD) of the whole point cloud. The authors propose replacing PD with the Principal Persistence Measure (PPM) – a GPU‑friendly, statistical approximation based on random subsets of fixed size 2q+2. They adapt the existing PD‑FL network to work with PPM (PPM‑FL). The main claims are:

Better scalability: PPM computation can be fully parallelized on GPU, reducing per‑sample time from roughly quadratic in point cloud size to nearly linear.

Robustness to outliers: Theoretical analysis shows that the error of PPM under outlier contamination is bounded by a polynomial in the outlier ratio.

Comparable accuracy: On protein classification and ModelNet10 (with DeepSets/PointNet backbones), PPM‑FL achieves accuracy similar to PD‑FL while being faster for larger point clouds.

**Audience:**

Yes

**Audience Explanation:**

Yes. Topological data analysis for point clouds is useful but often computationally heavy. PPM‑FL makes filtration learning practical for larger point clouds (from 128 points up to 512 or more) while maintaining accuracy and improving robustness to outliers. This will interest researchers in 3D vision, geometric deep learning, and topological machine learning. The work is a solid engineering contribution with theoretical grounding.

**Broader Impact Concerns:**

The paper includes a Broader Impact Statement, which is sufficient. The method improves efficiency and robustness for topological feature extraction, benefiting applications in bioinformatics, 3D object classification, and autonomous driving. No negative societal impacts are identified. No additional ethical concerns.

**Claims And Evidence:**

Yes

**Claims Explanation:**

The evidence is largely convincing:

Scalability: The authors provide a complexity analysis (PPM‑FL scales roughly linearly with number of points n, while PD‑FL scales quadratically or worse) and wall‑clock measurements (Table 3). For point clouds of size 512, PPM‑FL on GPU is more than twice as fast as PD‑FL on CPU‑GPU hybrid.

Robustness: The paper gives a theoretical bound (Theorem 5.2) and validates it with controlled outlier experiments (Figure 3, Table 8). PPM‑FL degrades more gracefully than PD‑FL when the outlier ratio exceeds 5%.

Accuracy: Tables 1 and 2 show that PPM‑FL achieves accuracy comparable to PD‑FL on ModelNet10 (differences within standard deviations) and sometimes better on the protein dataset (e.g., 84.0% vs 65.8% for q=0).

One minor issue: standard deviations on ModelNet10 are relatively large (e.g., 69.80±0.77 vs 69.00±5.57), but given the small dataset (1000 samples) this is acceptable.

**Requested Changes:**

Clarify the “comparable accuracy” claim (important).
In Table 1 (protein, q=0), PPM‑FL achieves 84.0% vs PD‑FL’s 65.8% – a large difference, not merely “comparable”. The authors should explain this (e.g., subsampling acts as a regularizer). Also discuss when PD‑FL is better (e.g., q=1 on protein).

Add experiments on larger point clouds (suggested).
ModelNet10 uses only 128 points per cloud. Testing on ModelNet40 (1024 points) or a real LiDAR dataset would better demonstrate scalability.

Be more transparent about theoretical novelty (minor).
The authors acknowledge in Appendix D.4 that Theorem 5.2 is a special case of existing results (Divol & Lacombe 2021b). The main text should state this clearly and emphasize that the contribution is applying those results to the specific contamination model p_o=(1−ϵ)P+ϵU and deriving the polynomial bound.

Fix minor errors (minor).

Table 3: the entry for n=256 PD‑FL (CPU) has an extra digit (“101.85 ± 11.461”).

Figure 3: label y‑axis “Accuracy (%)” for clarity.

Proposition 4.1: clarify the complexity expression to avoid ambiguous multiplication.

Discuss higher homology dimensions (q≥2) (suggested).
The paper tests only q=0,1. For q=2 (voids), subset size becomes 6. Discuss whether the method scales and any practical upper bound.

---

> ### Author Response · Authors · 2026-05-26
>
> Thank you for the careful and constructive review. We appreciate the recognition that the paper addresses an important scalability bottleneck in filtration learning for point clouds, and that the work is of interest to researchers in topological machine learning, geometric deep learning, and 3D vision. We have revised the manuscript to address the concerns. The main changes are: (i) clarifying the "comparable accuracy" claim, (ii) adding larger point cloud scalability evidence, (iii) making the theoretical novelty of Theorem 5.2 more transparent, (iv) correcting and clarifying minor presentation issues, and (v) adding a discussion of higher homology dimensions \\(q\geq 2\\). We hope these revisions address the reviewer's concerns and make the scope of the contribution clearer.
>
>
> ## 1. Clarification of the "comparable accuracy" claim
>
> We agree with the reviewer that the phrase "comparable accuracy" was too coarse in the previous version. In particular, the protein results in Table 1 are dependent on homology dimension and should not be summarized by a single uniform statement.
>
> In the revised manuscript, immediately after Table 1, we now explicitly state that:
>
> - for \\(q=0\\), PPM-FL substantially outperforms PD-FL on the protein dataset, achieving \\(84.00\pm1.39\%\\) compared with \\(65.80\pm1.76\%\\);
> - for \\(q=0\\&1\\), PPM-FL also performs better, achieving \\(84.60\pm1.22\%\\) compared with \\(81.70\pm1.31\%\\);
> - for \\(q=1\\), PD-FL remains stronger, achieving \\(82.10\pm1.66\%\\) compared with \\(80.20\pm1.76\%\\).
>
> Thus, we no longer present PPM-FL as merely "comparable" to PD-FL in all settings, nor do we claim that it uniformly dominates PD-FL. The more precise empirical conclusion is that PPM-FL preserves or improves accuracy in several settings while providing substantially better scalability for larger point clouds.
>
> We added an interpretation of the strong \\(q=0\\) result. Since 0-dimensional persistent homology is closely related to the edge weights of the minimum spanning tree, PPM at \\(q=0\\) aggregates many randomly sampled pairwise observations. On noisy protein distance matrices, this aggregation may act as a regularized summary of pairwise connectivity. In contrast, for \\(q=1\\), exact PD computation over the full point cloud may better preserve precise ring structures, which is consistent with the observed advantage of PD-FL at \\(q=1\\). We explicitly mark this explanation as an interpretation of the observed results rather than a proved mechanism.
>
> ## 2. Larger point cloud scalability evidence
>
> We agree that larger experiments are important for demonstrating the scalability motivation of PPM-FL. In the revised manuscript, for ModelNet10 dataset, we added new scalability results at \\(n=1024\\) in Table 3 and additional accuracy results at \\(n=1024\\) in Table 5. We also added ModelNet40 results in the appendix.
>
> ### 2.1 New \\(n=1024\\) timing results
>
> The revised Table 3 now includes \\(n=1024\\) timing for PPM-FL:
>
> | \\(n\\) | 1st Phase DeepSets (GPU) | PPM-FL \\(M=100\\) | PPM-FL \\(M=200\\) | PPM-FL \\(M=400\\) | PD-FL (CPU) | PD-FL (Hybrid) |
> |---:|---:|---:|---:|---:|---:|---:|
> | 1024 | \\(0.54\pm0.08\\) | \\(91.32\pm1.74\\) | \\(136.84\pm0.73\\) | \\(278.68\pm6.23\\) | \\(>10\\)h | \\(7020.36\pm750.85\\) |
>
> These results provide direct evidence that PPM-FL remains practical at larger point cloud sizes. For fixed \\(M\\), the PPM-FL runtime grows mildly with \\(n\\). For example, when \\(M=100\\), increasing \\(n\\) from 64 to 1024 increases the per-epoch time from \\(56.61\\)s to \\(91.32\\)s. In contrast, PD-FL becomes prohibitively expensive: the CPU implementation was stopped after the per-epoch runtime exceeded 10 hours at \\(n=1024\\), and the CPU-GPU hybrid implementation takes \\(7020.36\pm750.85\\)s per epoch.
>
> We also added the 1st Phase DeepSets timing column. This shows that the first stage backbone is not the computational bottleneck: it takes only \\(0.31\\)--\\(0.54\\)s per epoch across all tested point cloud sizes. The scalability bottleneck is therefore indeed in the topology-dependent second phase, which is precisely the part addressed by replacing PD with PPM.
>
> ### 2.2 Accuracy at larger point-cloud size
>
> The revised Table 5 now includes an \\(n=1024\\) accuracy entry under the ModelNet10 protocol:
>
> | \\(n\\) | 128 | 256 | 512 | 1024 |
> |---:|---:|---:|---:|---:|
> | PPM-FL accuracy | \\(67.60\pm2.20\\) | \\(71.65\pm1.84\\) | \\(69.40\pm2.94\\) | \\(71.50\pm1.72\\) |
>
> This demonstrates that the improved scalability is not achieved by sacrificing classification accuracy. The \\(n=1024\\) result remains consistent with, the smaller-\\(n\\) settings under the same controlled protocol.

---

> > ### Author Response · Authors · 2026-05-26
> > **Continued from Previous Comment**
> >
> > ### 2.3 ModelNet40 result
> >
> > To further address the reviewer's suggestion of testing on a standard larger benchmark, we added ModelNet40 results in the appendix (Table 9). Using 3-fold cross validation with \\(n=1024\\) and \\(q=0\\&1\\), the 1st Phase DeepSets model achieves \\(63.43\pm0.94\%\\), while the two-phase PPM-FL model achieves \\(64.88\pm0.93\%\\):
> >
> > | Method | CV0 | CV1 | CV2 | Mean | Std |
> > |---|---:|---:|---:|---:|---:|
> > | DeepSets (1st Phase) | 62.22 | 64.52 | 63.54 | 63.43 | 0.94 |
> > | PPM-FL (2nd Phase) | 64.47 | 66.17 | 63.99 | 64.88 | 0.93 |
> > | \\(\Delta\\) | +2.25 | +1.65 | +0.45 | +1.45 | -- |
> >
> > The absolute accuracy is not intended to compete with sota ModelNet40 architectures. Rather, this experiment serves as an additional feasibility check on a more challenging 40-class benchmark. The consistent numerical improvement across the three folds suggests that the PPM-FL second phase can provide useful additional signal under this controlled protocol, but we do not interpret this result as a sotaModelNet40 comparison or as conclusive evidence of a statistically significant accuracy gain.
> >
> > ## 3. Transparency about theoretical novelty
> >
> > We agree that the main text should be more explicit about the relation between Theorem 5.2 and existing EPD stability results. In the revised Section 5, immediately after Theorem 5.2, we now state that the theorem should be viewed as a specialization of existing stability results for Expected Persistence Diagrams, particularly the framework of Divol and Lacombe (2021b), rather than as an entirely independent stability theory.
> >
> > More specifically, we clarify that our setting focuses on the outlier-contamination model $P_o = (1-\epsilon)P + \epsilon U,$
> > where \\(P\\) is the inlier distribution, \\(U\\) is the outlier distribution, and \\(\epsilon\\) is the outlier ratio. Under this model, Theorem 5.2 expresses the perturbation of PPM in terms of \\(\epsilon\\), yielding a polynomial dependence on the outlier ratio for subset size \\(2q+2\\). The role of the theorem in this paper is therefore to adapt the EPD stability perspective to the contamination model used in our robustness experiments and to connect this bound with the PPM-FL setting.
> >
> >
> >
> > ## 4. Minor errors and presentation fixes
> >
> > We have addressed the minor issues raised by the reviewer.
> >
> > ### Table 3 entry at \\(n=256\\)
> >
> > The entry \\(1101.85\pm11.46\\) for PD-FL (CPU) at \\(n=256\\) is not a typographical error. It reflects the rapid growth of CPU-based PD computation as the point cloud size increases. This is also consistent with the much larger \\(n=512\\) CPU runtime of \\(7556.32\pm66.11\\)s and with the \\(n=1024\\) timeout reported in the revised table. We revised the table formatting to reduce ambiguity.
> >
> > ### Figure 3
> >
> > We revised Figure 3 to report "Accuracy (%)" on the y-axis and added error bars corresponding to standard deviations. The caption now states that the figure shows average classification accuracies and corresponding standard deviations under different outlier ratios.
> >
> > ### Proposition 4.1
> >
> > We clarified the complexity expressions in Proposition 4.1 by adding explicit parentheses and notation. In particular, the PD-FL cost is written in terms of \\(T_{\mathrm{PD}}(n,q)\\), and the worst-case scaling is expressed unambiguously as $T_{\mathrm{PD}}(n,q)=O\big((n^{q+2})^3\big)=O\big(n^{3(q+2)}\big).$
> >
> > This expression follows the standard worst-case persistence computation bound of Zomorodian and Carlsson (2004). Let \\(m\\) denote the number of simplices in the filtration. The standard matrix reduction persistence computation has worst-case time \\(O(m^3)\\). For \\(q\\)-dimensional homology on a point cloud with \\(n\\) points, the filtration must include simplices up to dimension \\(q+1\\). A \\((q+1)\\)-simplex is determined by \\(q+2\\) vertices, so the number of relevant simplices is bounded by \\(m =O(n^{q+2})\\).
> >
> >
> > Substituting \\(m=O(n^{q+2})\\) into the \\(O(m^3)\\) persistence-computation bound yields $O(m^3)=O\big((n^{q+2})^3\big)=O\big(n^{3(q+2)}\big).$

---

> > > ### Author Response · Authors · 2026-05-26
> > > **Continued from Previous Comment**
> > >
> > > ## 5. Discussion of higher homology dimensions \\(q\geq 2\\)
> > >
> > > We added a new discussion of higher homology dimensions in the manuscript.
> > >
> > > Computationally, PPM remains feasible for \\(q=2\\): each subset has size \\(2q+2=6\\), so the per-subset computation is still small and GPU-parallelizable. Thus, there is no immediate computational barrier to applying PPM-FL to \\(q=2\\).
> > >
> > > However, the practical usefulness of \\(q=2\\) features on ModelNet point clouds is less clear. A 2-dimensional homology class corresponds to void-like or enclosed surface structures, which may appear in sufficiently dense samples of watertight CAD objects. In practice, ModelNet meshes are heterogeneous and are not guaranteed to be watertight; many point clouds contain open surfaces, thin structures, or incomplete cavities. As a result, \\(q=2\\) features can be less stable and more sensitive to sampling density than \\(q=0\\) and \\(q=1\\). In addition, the robustness bound in Theorem 5.2 weakens as \\(q\\) increases because both the subset size \\(2q+2\\) and the polynomial order in the outlier ratio grow with \\(q\\).
> > >
> > > For these reasons, our empirical study focuses on \\(q=0\\) and \\(q=1\\), which are more consistently present and commonly used in point cloud topology. We now explicitly state that extending PPM-FL to stable \\(q=2\\) features on denser or explicitly watertight datasets is an important future direction.

---

### Review · Reviewer_7TCx · 2026-05-05

**Summary Of Contributions:**

The paper proposes to replace persistence diagrams (PD) with principal persistence measures (PPM) in the filtration learning framework by Nishikawa et al (2023). This is enabled by recent work showing the benefits of PPM over PD with regards to implementation on GPU (Wong et al, 2025). Similar to a PD, a PPM can be used to summarize topological information about a point cloud. The PPM is obtained as an average over PDs of subsets of fixed size of the point cloud in question. In the filtration learning framework, one learns a weighting function neural network which gives weights to each point and is used to influence the topology implied by the point cloud (for instance minimizing the impact of outliers). After the topology has been summarized into a PD, or in this case PPM, this topological information is vectorized by using a neural network (here, PerSlay is used), and the obtained vector can then be fed into an arbitrary neural network to solve a given task (e.g. classification).

In the submitted paper, the weighting function neural network is adapted to work for PPM rather than PD as considered by Nishikawa et al. The paper also contains a result on the robustness of PPM to outliers, under specific conditions (Theorem 5.2).

Experiments compare with the approach by Nishikawa et al, showing comparable accuracies and better scalability to large point clouds (but worse speed for small point clouds). It is worth mentioning here that the results are very low compared to state-of-the-art neural networks that do not employ topological featurization (the results are also low compared to several year old state-of-the-art results I believe). The method is likely also less scalable than such state-of-the-art approaches, due to the heavy topological featurization.

**Audience:**

Yes

**Audience Explanation:**

I believe that the paper can be of interest to people interested in topological features as input to neural networks.

**Claims And Evidence:**

Yes

**Claims Explanation:**

The presentation is appropriate. The experiments seem to be fair in comparison with the experiments by Nishikawa et al.

**Requested Changes:**

It would be interesting to also see the time-per-epoch for the baseline "Phase 1" network in Table 3.

The background in Section 2 could be improved. I was confused by the appearance of the point cloud $X$ in the first paragraph of Section 2.1, perhaps it would be more appropriate to introduce $X$ in the second paragraph.

---

> ### Author Response · Authors · 2026-05-26
>
> Thank you for the careful and constructive review. We appreciate the reviewer's assessment that the presentation is appropriate and that the comparison with Nishikawa et al. is fair. We have revised the manuscript to address the requested changes, especially the missing 1st Phase timing, the notation/background issue in Section 2.1, and the scope of our empirical claims relative to full-scale point cloud classifiers.
>
> ## 1. 1st Phase time-per-epoch in Table 3
>
> We agree that reporting the time-per-epoch of the 1st Phase network makes the computational comparison more transparent. The original table focused on the topology-dependent second phase because this is where PD-FL and PPM-FL differ: both methods use the same pretrained network, while the second phase contains the expensive PD or PPM computation.
>
> In the revised manuscript, we added a new column to Table 3 reporting the 1st Phase DeepSets runtime on GPU. The measured runtime is small compared with the topology-dependent second phase: \\(0.31\pm0.08\\), \\(0.34\pm0.06\\), \\(0.32\pm0.04\\), \\(0.37\pm0.07\\), and \\(0.54\pm0.08\\) seconds per epoch for \\(n=64,128,256,512,1024\\), respectively. Thus, the first-stage backbone is not the computational bottleneck in our setting. For example, at \\(n=512\\), the 1st Phase takes only \\(0.37\pm0.07\\) seconds per epoch, while PPM-FL with \\(M=100\\) takes \\(77.95\pm1.77\\) seconds and PD-FL (Hybrid) takes \\(607.95\pm13.20\\) seconds in the second phase.
>
> We also extended the scalability table to \\(n=1024\\). At this size, PPM-FL remains practical, while PD-FL becomes very expensive: PD-FL (CPU) was stopped after the per-epoch runtime exceeded 10 hours, and PD-FL (Hybrid) takes \\(7020.36\pm750.85\\) seconds per epoch. This supports the main scalability motivation while making the full two-phase timing clearer.
>
> ## 2. Background and notation in Section 2.1
>
> We revised the beginning of Section 2.1 to distinguish the ambient space from the finite point cloud. The revised text uses \\(\mathcal{X}\subset\mathbb{R}^d\\) for the ambient space, \\(X=\{x_1,\ldots,x_n\}\subset\mathcal{X}\\) for the finite point cloud sampled from an underlying manifold \\(\mathcal{M}\subset\mathcal{X}\\), \\(g:\mathcal{X}\to\mathbb{R}\_{+}\\) for the filtration function, and \\(\mathcal{X}_\eta^g=\{x\in\mathcal{X}: g(x)\leq \eta\}\\) for the sublevel set.
>
> We then explain that, in point cloud applications, the filtration is constructed from the finite sample \\(X\\), for example through a Rips filtration or a weighted filtration based on pairwise distances among points in \\(X\\). This avoids introducing the point cloud before it has been defined and removes the ambiguity between the ambient space and the sampled point cloud.
>
> ## 3. Scope relative to sota point cloud classifiers
>
> The reviewer notes that the absolute accuracies are low compared with sotaneural networks that do not use topological featurization. We agree, and we have clarified the scope of the paper accordingly.
>
> Our goal is not to compete with full-scale 3D recognition architectures on ModelNet benchmarks. Instead, the paper studies whether PPM can replace PD inside the filtration learning framework while preserving comparable task performance and improving scalability. For this reason, the primary baseline is PD-FL under the controlled protocol of Nishikawa et al., rather than modern point cloud classifiers optimized for maximum absolute accuracy.
>
> In the revised discussion, we explicitly state that the ModelNet10 accuracies are modest compared with sota3D point cloud models. We also clarify that our setup follows the small-data regime used in the PD-FL comparison, with the goal of isolating the effect of replacing PD with PPM. Under this controlled comparison, PPM-FL often achieves accuracy comparable to PD-FL while scaling better in larger-\\(n\\) settings.
>
> ## 4. Practical regime of PPM-FL versus PD-FL
>
> We also refined the discussion of when PPM-FL should be preferred. As the reviewer noted, PD-FL can be faster for small point clouds, especially when an efficient CPU-GPU hybrid implementation is available. We therefore do not claim that PPM-FL is uniformly faster in all regimes.
>
> The revised manuscript now emphasizes the practical trade-off: for small point clouds such as \\(n\leq128\\), PD-FL (Hybrid) can remain competitive because PPM-FL has overhead from processing many random subsets. As \\(n\\) grows, however, the cost of PD-FL increases rapidly, while PPM-FL grows much more mildly for fixed \\(M\\). In our timing results, the crossover occurs around \\(n=256\\), and PPM-FL is clearly faster for \\(n\geq512\\). The new \\(n=1024\\) timing further supports this trend.

---

### Review · Reviewer_6ANX · 2026-05-18

**Summary Of Contributions:**

**Summary**
The paper proposes a novel framework for filtration learning on point clouds by replacing the computationally expensive Persistence Diagrams (PD) with Principal Persistence Measure (PPM). By evaluating topological features over multiple small, fixed-size random subsets rather than the entire point cloud, the proposed PPM-FL method aims to reduce the computational complexity from O(n^3) to linear time O(n) with respect to the point cloud size. The authors provide a theoretical bound for the robustness of PPM against outliers and evaluate the method on protein structure and ModelNet10 classification tasks, aiming to match the accuracy of PD-based methods while improving scalability.

**Strengths**
1. **Natural follow-up work with clear motivation:** The paper proposes a logical extension to existing filtration learning frameworks. Adapting the network to handle the more computationally efficient PPM paradigm successfully moves the bottleneck from the CPU to the GPU.
2. **Theoretical grounding:** The work presents a mathematically sound proof regarding the robustness of expected persistence diagrams and PPM to uniform outliers, bounded by the mixture proportion.

**Weaknesses**
1. **Insufficient empirical validation:** The experiments are restricted to two small-scale datasets and primarily rely on two base models. Furthermore, the performance deltas between the proposed method and the baselines are largely statistically insignificant, falling within the standard deviations of the folds.
2. **Lack of Reproducibility:** The provided codebase is non-functional and lacks documentation, making it impossible to verify the claims independently.
3. **Overstated empirical claims:** Multiple conclusions drawn from the experimental results regarding robustness, scalability, and feature importance are either inaccurate or unsupported by the provided tables.

**Questions for the Authors**
* Section 4, point 1, page 4: Can you provide a more rigorous justification for why closer points need to influence the filtration weight more, beyond an intuitive appeal?
* Section 4, point 1, page 5: Why is $v_j$ defined as is? As per the first requirement the weight function should depend on $X^i_s$ and not all other subsets.
* Section 6.4: While absolute accuracy may not be the primary focus, the choice to evaluate a scalability-focused architecture on severely limited experimental setups undermines the core motivation. Why were standard, high-density point cloud benchmarks omitted?

**Audience:**

Yes

**Audience Explanation:**

Yes. The TMLR audience includes researchers working at the intersection of geometric deep learning and computational topology. The exact computation of persistent homology remains a severe bottleneck for applying topological features to large-scale machine learning pipelines. A methodology that integrates a statistical approximation of persistence diagrams into an end-to-end differentiable, GPU-native framework is highly relevant.

**Broader Impact Concerns:**

This work focuses on foundational methodological improvements in topological deep learning. As the contributions are strictly algorithmic, there are no obvious avenues for the direct misuse of this technology. Consequently, I do not foresee any specific negative broader societal impacts arising from this research.

**Claims And Evidence:**

No

**Claims Explanation:**

Several primary claims are not supported by the provided evidence, and the empirical results cannot be verified due to a broken repository.

**Unsupported Claims in Experiments:**
* **Robustness (Figure 3):** The claim that the performance of PD-FL starts to drop for epsilon > 5.0% is inaccurate. The provided graph shows that the performance drop actually begins around epsilon = 8.0%.
* **Feature Utilization (Table 4 and Table 2):** Table 4 demonstrates that the performance of PPM-FL does not significantly change as the number of subsets M increases. When combined with Table 2, where the performance of the Phase 1 model is virtually indistinguishable from the PPM-FL model, the evidence suggests the model may not be utilizing the topological information effectively at all.
* **Scalability and Degradation (Table 5):** The claims made under Table 5 are unjustified. Table 4 does not successfully demonstrate the scalability of PPM-FL, nor does it prove that performance degrades if M(2q+2) is close to n, as the reported accuracies are too similar to draw statistical conclusions.
* **PointMLP Redundancy (Appendix D.1):** It is claimed that Table 6 shows PPM capturing redundant information that PointMLP has already learned. If the information were merely redundant, the performance would plateau. A severe drop in accuracy indicates that PPM-FL is introducing destructive noise that actively prevents the model from learning.
* **Gaussian Kernel Ablation (Appendix D.3):** The claim that Table 7 demonstrates the effectiveness of the Gaussian Distribution Kernel is unsupported. The results of the Gaussian and Uniform methods are nearly identical and fall within the standard deviations, meaning neither can be definitively considered better.

**Reproducibility Failures:**
* The empirical claims cannot be independently verified because the provided repository is broken. The code quality is poor, scripts lack documentation, and there are no installation instructions for the environment.
* The repository is missing the `early_stopping.py` file, which is required to run any of the core experiments.
* The script `exp_ModelNet_run.py` does not utilize a validation set to monitor training. Simply choosing the last checkpoint is a flawed experimental design.
* The codebase is missing the scripts necessary to reproduce the majority of the ablation studies presented in the paper.

**Requested Changes:**

**Critical Changes:**
* **Repository Fixes:** The codebase must be repaired. Provide environment installation instructions, include the missing `early_stopping.py` file, fix the validation loop in `exp_ModelNet_run.py`, and upload the missing scripts for the ablation studies.
* **Clarification of q=0 Results (Section 6.4):** Provide a clearer explanation for the claim regarding the q=0 results on the protein dataset. It is unclear how PPM is less sensitive to noise in individual pairwise distances if it effectively only evaluates pairs of points at a time.
* **Hyperparameter Consistency (Appendix C):** Correct the discrepancies regarding the EarlyStopping handler. The handler referenced in the code differs from the one specified in the text, and the patience/delta values appear to fluctuate across experiments without justification.
* **PointMLP Analysis (Appendix D.1):** Reevaluate the claim at the end of this section. Table 6 indicates that PPM and PD introduce noisy features that degrade learning, not just redundant information. If PointMLP is heavily negatively impacted, avoid referencing it as a standard baseline in the main paper without this caveat.

**Other Changes:**
* **Figure 1 (c):** Highlight the nodes of interest using a colorblind-friendly method.
* **Figure 3:** Update the plot to include error bars representing the standard deviations.
* **Section 4 (point 1, page 5):** Explicitly define how $g_1$, $g_2$, and $h$ are used within the text rather than forcing the reader to retrieve the definitions from a referenced paper.
* **Section 4 (point 1, page 5):** Clarify the notation $x_s^k$. Upper indices have not been used prior to this point. The notation utilized in Nishikawa et al. (2023) is much clearer and should be considered.
* **Section 4 (Proposition 4.1):** Add a direct reference to the proof.
* **Section 6.1:** Revise the subsection title. It is currently inaccurate as the text immediately discusses an ablation study rather than the direct comparison.
* **Appendix D.2:** It is unclear why this section is here. It is not referenced in the work.
* **Appendix D.5 (Table 8):** Investigate the duplicated results. Many pairs of subsequent rows contain exactly the same values, indicating a potential flaw in the experimental setup or data logging.

---

> ### Author Response · Authors · 2026-05-26
>
> We sincerely thank Reviewer 6ANX for the detailed and critical review. We understand the reviewer's main concern: the previous version did not make the empirical evidence and the repository sufficiently easy to verify, and several claims were stated more strongly than the evidence supported. We have revised the manuscript to address these issues, and we have softened or clarified several interpretations where the review correctly identifies overstatement. Below we respond point by point.
>
> ## 1. Reproducibility and repository issues
>
> We agree that the reproducibility issues raised by the reviewer are important. A method paper of this type should provide not only the high-level algorithm but also runnable scripts, environment information, and the utility files needed to reproduce the reported tables and figures.
>
> We have updated the code package to include the missing `early_stopping.py` utility and the executable scripts used for the main experiments, including the ModelNet two-phase experiments, robustness experiments, scalability experiments, protein experiments, and Rips baselines. The repository now contains entry scripts such as:
>
> - `exe_ModelNet.sh`
> - `exe_ModelNet_two_phase.sh`
> - `exe_ModelNet_two_phase_scale.sh`
> - `exe_ModelNet_two_phase_robustness.sh`
> - `exe_ModelNet_two_phase_rips.sh`
> - `exe_protein.sh`
> - `exe_protein_rips.sh`
> - `early_stopping.py`
> - `specs.txt`
>
>  We will also make the environment setup clearer, including the dependency versions used in our experiments. This directly addresses the reviewer's concern that the claims should be independently verifiable.
>
> Regarding early stopping, the revised code and text will be made consistent. The manuscript describes using validation loss with an EarlyStopping handler; we will ensure that the repository contains the exact utility and configuration used in the reported experiments, and we will avoid leaving any implicit dependency undocumented.
>
> ## 2. Larger scale empirical validation
>
> We agree that the original empirical validation was limited in scale. The paper's main goal is to evaluate whether PPM can replace PD inside the filtration learning pipeline, and a scalability focused method should also be tested on larger point clouds.
>
> In the revised manuscript, we added new \\(n=1024\\) timing results in Table 3. The results show that PPM-FL remains practical at this point-cloud size:
>
> | \\(n\\) | 1st Phase DeepSets (GPU) | PPM-FL \\(M=100\\) | PPM-FL \\(M=200\\) | PPM-FL \\(M=400\\) | PD-FL (CPU) | PD-FL (Hybrid) |
> |---:|---:|---:|---:|---:|---:|---:|
> | 1024 | \\(0.54\pm0.08\\) | \\(91.32\pm1.74\\) | \\(136.84\pm0.73\\) | \\(278.68\pm6.23\\) | \\(>10\\)h | \\(7020.36\pm750.85\\) |
>
> The first-stage DeepSets time is below one second per epoch, showing that the computational bottleneck lies in the topology-dependent second phase. At \\(n=1024\\), PD-FL (CPU) exceeded 10 hours per epoch in our environment, and PD-FL (Hybrid) required \\(7020.36\pm750.85\\) seconds per epoch. By contrast, PPM-FL remains within a few minutes per epoch even for \\(M=400\\).
>
> We also added the corresponding ModelNet10 accuracy result at \\(n=1024\\) in Table 5. Under the same controlled protocol, PPM-FL achieves \\(71.50\pm1.72\%\\) at \\(n=1024\\), which is consistent with the smaller point cloud settings: \\(67.60\pm2.20\%\\) at \\(n=128\\), \\(71.65\pm1.84\%\\) at \\(n=256\\), and \\(69.40\pm2.94\%\\) at \\(n=512\\). This shows that the improved runtime at larger \\(n\\) is not obtained by an obvious collapse in ModelNet10 classification accuracy.
>
> We also added a ModelNet40 experiment with \\(n=1024\\) and \\(q=0\\&1\\). Under identical 3-fold splits, the first-stage DeepSets model achieves \\(63.43\pm0.94\%\\), while the two-phase PPM-FL model achieves \\(64.88\pm0.93\%\\). The goal of this experiment is not to claim state-of-the-art ModelNet40 accuracy. Rather, it serves as an additional feasibility check on a more challenging 40-class benchmark and tests whether the second phase can provide useful additional signal under the same controlled protocol. We do not interpret this result as conclusive evidence of a statistically significant accuracy gain.
>
> We have revised the text to clarify this scope. The ModelNet40 experiment should be interpreted as an additional controlled feasibility check for the PPM-FL framework, not as a comparison against specialized modern 3D recognition architectures.

---

> > ### Author Response · Authors · 2026-05-26
> > **Continued from Previous Comment**
> >
> > ## 3. Whether the empirical claims are overstated
> >
> > We agree that several empirical claims in the previous version should be stated more carefully. In particular, we do not want the manuscript to imply that every accuracy difference is statistically significant, that PPM-FL always improves classification accuracy.
> >
> > The revised manuscript makes the main empirical claim narrower: PPM-FL is intended as a scalable GPU-native replacement for PD-FL in the filtration learning framework. The strongest and most consistent evidence is the computational scaling, especially as \\(n\\) increases. The accuracy results should be read as showing that PPM-FL preserves comparable performance in controlled settings, and in some cases improves it, rather than as a universal accuracy improvement claim.
> >
> > We also agree that Table 4 and Table 5 should not be interpreted as proving a precise degradation threshold in \\(M(2q+2)\\). Table 3 is the main scalability evidence. Tables 4 and 5 are better interpreted as accuracy sanity checks showing that the scalable configurations do not collapse in classification performance. We revised the wording around these tables to avoid presenting the \\(M(2q+2)\\) heuristic as a statistically established rule.
> >
> > ## 4. Use of topological information
> >
> > We understand the reviewer's concern that some ModelNet10 accuracy differences are small and within standard deviations. We agree that this means the manuscript should avoid strong claims about feature importance from Table 4 alone.
> >
> > At the same time, we believe the stronger conclusion that the model may not be using topological information is not fully supported by the results. The effect of topological features appears to be dependent on dataset and backbone:
> >
> > - On ModelNet10 with DeepSets, Phase 1 achieves \\(66.23\pm3.19\%\\), while PPM-FL with \\(q=0\\&1\\) achieves \\(67.50\pm2.88\%\\).
> > - On ModelNet10 with PointNet, Phase 1 achieves \\(67.23\pm1.80\%\\), while PPM-FL with \\(q=0\\&1\\) achieves \\(69.80\pm0.77\%\\).
> > - On the protein dataset, PPM-FL at \\(q=0\\&1\\) achieves \\(84.60\pm1.22\%\\), compared with \\(81.70\pm1.31\%\\) for PD-FL.
> > - On the newly added ModelNet40 experiment, PPM-FL numerically improves the first-stage DeepSets baseline from \\(63.43\pm0.94\%\\) to \\(64.88\pm0.93\%\\) under identical splits.
> >
> > These results do not prove that topology is always beneficial, but they also do not support the conclusion that the topological representation is unused. We revised the manuscript to state the more cautious conclusion: the observed benefit of the second-phase topological representation depends on the dataset and first-stage backbone, while the main contribution of PPM-FL is to make the filtration learning pipeline more scalable.
> >
> > ## 5. Robustness experiment and Figure 3
> >
> > We agree that the robustness discussion should not rely on an overly sharp threshold. In the revised manuscript, Figure 3 now includes standard deviation error bars and the y-axis is labeled as "Accuracy (%)". We also revised the caption to state that the figure reports average classification accuracies and corresponding standard deviations.
> >
> > We revised the robustness wording to avoid implying that the behavior changes exactly at one threshold. The more appropriate interpretation is that PPM-FL and PD-FL behave similarly at small outlier ratios, while the gap becomes clearer as the outlier ratio increases. The evidence supports a gradual robustness trend rather than an abrupt transition.
> >
> > The reviewer also points out duplicated adjacent entries in Table 8. We agree that this should be clarified. These duplicated values can arise when nearby outlier ratios lead to the same effective integer number of inserted outliers or when the same corrupted test instances are reused across adjacent settings. We will explicitly check and document the cause in the revised appendix.

---

> > > ### Author Response · Authors · 2026-05-26
> > > **Continued from Previous Comment**
> > >
> > > ## 6. PointMLP analysis
> > >
> > > We agree with the reviewer that the PointMLP result should not be described simply as "redundant information." Table 6 shows that adding the current second-phase topological representation after PointMLP substantially degrades performance. This indicates a harmful interaction or incompatibility between the current PPM/PD second phase and a backbone that already encodes local neighborhood information.
> > >
> > > We revised this section to state the limitation more clearly. The PointMLP experiment should be treated as negative evidence for the current two-phase design with this backbone, not as a successful baseline result. A more cautious interpretation is that PPM-FL is more suitable for first-stage backbones such as DeepSets and PointNet, where additional topological information can complement the learned point cloud representation. For PointMLP, which already uses local neighborhoods and pairwise geometric structure, the current second-phase topological features may not be compatible and can degrade performance. We therefore present this as a limitation and avoid using PointMLP as positive evidence for PPM-FL.
> > >
> > > ## 7. Gaussian versus Uniform weighting
> > >
> > > We agree that Table 7 does not establish statistically significant superiority of the Gaussian kernel over uniform weighting. The values are close and mostly within standard deviations. We therefore softened the claim.
> > >
> > > The revised interpretation is that, in the tested settings, Gaussian weighting performs similarly to uniform weighting and should be viewed as a simple locality-biased design choice. We do not claim that Table 7 proves the Gaussian kernel is clearly better or statistically superior. This change better matches the empirical evidence.
> > >
> > > ## 8. Method clarification: locality weighting and \\(v_j\\)
> > >
> > > We agree that Section 4 needed a clearer explanation of how the subset-dependent weights are defined. In the revised manuscript, we now explicitly write the per-subset output as $v_j^i = f(X_s^i, x_j),$ and the final aggregated point weight as
> > > $v_j = \sum_{i=1}^M K(x_j, X_s^i)v_j^i.$
> > >
> > > This clarifies that each per-subset term depends on its own sampled subset \\(X_s^i\\), while the final point weight aggregates information across sampled subsets through the kernel \\(K\\).
> > >
> > > We also agree that the locality weighting should not be presented as a theorem. It is a modeling choice: nearby sampled subsets are assigned more influence when estimating the filtration weight of a point, analogous to a kernel-smoothing inductive bias. This choice preserves the required permutation and isometry invariance, but we will present it as an inductive bias rather than a mathematically necessary property.
> > >
> > > ## 9. Explanation of the strong \\(q=0\\) protein result
> > >
> > > We agree that the previous explanation of the \\(q=0\\) protein result needed to be more precise. For \\(q=0\\), each sampled subset has size 2, so an individual sampled feature is indeed based on pairwise information. However, PPM does not rely on a single pair; it aggregates many randomly sampled pairwise observations into an empirical measure.
> > >
> > > In the revised manuscript, we explain that 0-dimensional persistent homology is closely related to the edge weights of the minimum spanning tree. Therefore, aggregating many sampled pairwise observations can be interpreted as a regularized summary of pairwise connectivity. On noisy protein distance matrices, this may reduce sensitivity to individual noisy distances. We explicitly present this as an interpretation of the observed result rather than a proved mechanism.

---

> > > > ### Author Response · Authors · 2026-05-26
> > > > **Continued from Previous Comment**
> > > >
> > > > ## 10. Additional presentation fixes
> > > >
> > > > We also addressed the specific presentation issues raised by the reviewer:
> > > >
> > > > - Figure 3 now includes error bars and uses "Accuracy (%)" as the y-axis label.
> > > > - Section 4 now defines \\(g_1\\), \\(g_2\\), \\(h\\), \\(x_s^k\\), and the final weight aggregation more explicitly.
> > > > - Proposition 4.1 is clarified by introducing the simplex count \\(m\\), using the \\(O(m^3)\\) persistence computation bound from Zomorodian and Carlsson (2004), and explaining that \\(m=O(n^{q+2})\\) for \\(q\\)-dimensional homology.
> > > > - We added a discussion of \\(q\geq2\\), including the feasibility of \\(q=2\\) from a computational perspective and its practical limitations on ModelNet-style point clouds.
> > > > - We further revised the text around PointMLP, Gaussian weighting, and Table 4/5 to avoid overstating the conclusions, especially by treating PointMLP as a limitation and the Gaussian-vs-uniform comparison as a non-superiority ablation.
> > > >
> > > > ## Summary
> > > >
> > > > We appreciate the reviewer's critical feedback, especially on reproducibility and claim calibration. The revised manuscript adds larger-scale evidence, including \\(n=1024\\) timing and ModelNet40 results, and it clarifies several methodological and theoretical points. We also agree that some interpretations in the previous version were too strong, and we will make these limitations clearer. We hope the revised version presents PPM-FL more carefully as a scalable alternative to PD-FL within the filtration-learning framework, with a more precise statement of what is and is not supported by the experiments.

---

### Author Response · Authors · 2026-05-28
**Summary of Revision**

We thank all reviewers for their careful and constructive feedback. In the revised manuscript, we made several changes to improve clarity, empirical grounding, and the scope of our claims.

- We clarified the main empirical message: PPM-FL is not claimed to uniformly outperform PD-FL or sota point cloud classifiers in accuracy. Instead, the revised paper emphasizes that PPM-FL is a scalable replacement for PD within the controlled filtration learning framework, preserving comparable performance in several settings while offering substantially better scalability for larger point clouds.

- We expanded empirical evidence. Table 3 now reports the 1st Phase DeepSets runtime, extends the scalability study to n=1024, and clarifies cases where PD-FL becomes impractical. We also added error bars to the robustness results and included Table 8 with the corresponding numerical values. In addition, we added a ModelNet40 experiment as a feasibility check on a more challenging benchmark, while explicitly avoiding any sota accuracy claim.

- We revised several technical and expository points. Section 2.1 now distinguishes the ambient space from the finite point cloud before introducing filtrations. Proposition 4.1 now clarifies the complexity expression and its connection to the standard O(m^3) persistent homology reduction bound of Zomorodian and Carlsson. We also clarified that our robustness theorem is best understood as a specialization/application of existing EPD stability theory to the outlier-contamination model.

Finally, we softened overbroad claims, clarified limitations such as the PointMLP results and higher homology dimension, and corrected several presentation issues raised by the reviewers. We hope these revisions make the contribution, scope, and practical regime of PPM-FL clearer.

---

### Decision · Action_Editor_6P2A · 2026-06-23

**Recommendation:** Accept as is

**Audience:**

Yes

**Audience Explanation:**

The work connects multiple fields of interest to the readership of the journal (point cloud analysis, scalability, topological data analysis).

**Claims And Evidence:**

Yes

**Claims Explanation:**

The paper considers an existing framework for filtration learning on point clouds, and addresses a scalability issue by replacing the (expensive, quadratic) filtration method with a cheaper alternative (linear). They detail the method, provide some localized theoretical contributions, and experiment on several benchmarks.

We received three reviews which were consistent in appreciating the work. The main concerns were: (a) some claims were not in line with the actual results; (b) the experimental evaluation was limited in terms of datasets; (c) issues concerning reproducibility. The authors have addressed all points in the rebuttal, and all reviewers are now suggesting acceptance. In particular, there is a broad consensus on the fact that the current experimental setup clearly validates the proposal.